# Optimizing the Combined Allocation of Land and Water to Agriculture in the Omo-Gibe River Basin Considering the Water-Energy-Food-Nexus and Environmental Constraints

**Sintayehu Legesse Gebre** [1,2,*]**, Jos Van Orshoven** [3] **and Dirk Cattrysse** [1]

[1] Center for Industrial Management, Department of Mechanical Engineering, Traffic & Infrastructure, KU Leuven (University of Leuven), Celestijnenlaan 300, 3001 Leuven, Belgium
[2] Department of Natural Resource Management, Jimma University, Jimma P.O. Box 378, Ethiopia
[3] Department of Earth and Environmental Sciences, Division of Forest, Nature and Landscape, KU Leuven (University of Leuven), Celestijnenlaan 200E, 3001 Leuven, Belgium
* Correspondence: sintayehulegesse@gmail.com or sintayehulegesse.gebre@kuleuven.be

**Abstract:** This study applied the Gebre optimization model to optimize the land and water usage in the Omo-Gibe river basin, Ethiopia, where competition among stakeholders and growing demands pose a challenge. This model was applied through a nexus approach to maximize benefits and minimize conflicting trade-offs. The main objective was to maximize the economic benefit from land and water allocation with the framework of the land-water-food-energy-environment nexus under climate change mitigation and river ecosystem services (LWFEEN). This model takes into account multiple dimensions, including economic, environmental, social, and technical factors, going beyond ordinary optimization models. It also incorporates an innovative crop succession allocation concept not often seen in the literature. This crop succession proposal includes sequences of cropping patterns and fallow land use options that closely resemble real-world farming practices. The results demonstrated that the Gebre optimization model effectively resolves the existing constraint conflicts and maximizes economic benefits by reducing costs, penalties, and environmental impacts, promoting sustainable use of natural resources in the Omo-Gibe river basin and avoiding conflicts among stakeholders. Therefore, this study offered decision-makers a strategic perspective on how to apply the Gebre-model within the context of the land-water-food-energy-environment nexus(LWFEEN) approach in river basins such as the Omo-Gibe, with the ultimate goal of achieving sustainable development.

**Keywords:** allocation; crop succession; Gebre-model; nexus; Omo-Gibe river basin; optimization

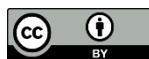

## 1. Introduction

Land and water resource allocation is a complex strategic and operational management challenge for decision-makers [1]. Hence it has a significant impact on local and regional socioeconomic development, ecosystem services, and political stability [2]. Recently, the competition for land and water resources has been growing to meet water, food, and energy demands [3,4]. Land and water are scarce natural resources, and their demand has increased due to population increase, urbanization, and industrialization. These led to exhaustive pressure on the existing limited natural resources [5,6]. Therefore, it is important to emphasize how to allocate land and water resources without compromising future generation and meets the current water-food-energy demands [7,8]. Hence, the key issue is where, for what purpose, and when to allocate land and water to achieve equity and satisfactory level for each land and water user. Failure to achieve a reasonable resource allocation will lead to instability and conflict among users [9,10]. So, by optimizing land and water allocation, the above challenges can be alleviated or reduced.

Mathematical programming approaches are frequently used to solve complex multi objectives problems [11]. Several mathematical decision-making tools have been developed and used in a different land and water resource allocation problems to address water-food-energy demands. Veintimilla et al. [12] used the MILP method to allocate river and reservoir water to meet spatio-temporal water demand in the Machángara River basin, Ecuador. They have considered only the water allocation part as a main component to meet WEF demands without considering land resources as an explicit requirement for food production. Deng et al. [13] employed a non-sorting genetic algorithm (NSGA) to optimize water resource allocation in the Han River basin in China. The study aimed to maximize economic efficiency and equity among water users. Many more studies have been done on water or land allocation to address the water-food-energy nexus [10,14,15]. The Gebre-model is the only outstanding mathematical model that considers the combined land and water resource allocation.

In this study, the main objective was to apply the **Gebre-model** to the Omo-Gibe river basin in Ethiopia. The Omo-Gibe river basin was selected because of the complexity and distribution of water resource development structures and its challenges to the land-water-food-energy-environment nexus. This issue called for an integrated and systematic approach to managing land and water resources to deliver ecosystem services. Furthermore, in this paper, we specifically addressed (a) the optimization of the allocation of the limited land resource to specific agricultural use types under environmental, economic, social, and technical constraints, (b) the optimization of the allocation of the limited and spatio-temporally variable water resources to different water users to meet water, food, energy, environmental conservation, and ecosystem demands, and (c) to maximize the overall net economic benefit from combined land and water allocation.

In general, this model contributed to the concept of integrated land and water resource planning and management as illustrated for the Omo-Gibe river basin. Moreover, it can be used as an optional decision support tool to address LWFEE nexus challenges and promote climate change mitigation and delivery of river ecosystem services (Figure 1).

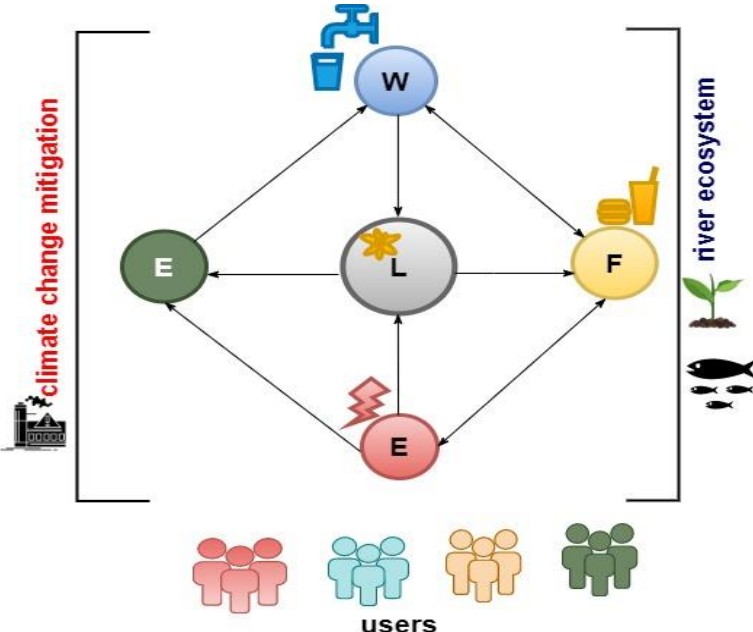

**Figure 1.** Schematic representation of the land-water-food-energy-environment nexus (LWFEEN) under climate change mitigation and river ecosystem service (*L—land; W—water; F—food; E—energy; E—environment*).

## 2. Materials and Methods

### 2.1. Description of the Study Area

The Omo-Gibe river basin is situated in southwestern Ethiopia along the central rift valley. The river starts from the northern highlands of the basin, flowing down to the southern lowland, fed by small and large tributaries [16]. The Omo-Gibe river is essentially one long river with different names in different regions. Indirectly, the Gibe river becomes the Omo river in the lowland area of the basin. It ultimately flows into Lake Turkana in Kenya. The basin has a 79,000 Km² area and approximately 849 km length from the north to the south end [17]. The Omo-Gibe river basin has significant importance to Ethiopia`s socio-economic development.

Currently, water resource development structures are present or being constructed in the basin. Three hydropower stations, namely Gibe_I, Gibe_II, and Gibe_III, are operational, with Gibe IV under construction. Both small and large-scale irrigation schemes (e.g., Kuraz irrigation farms) are present, while several attractive ecotourism sites remain untapped [17]. The model uses around 400 KM of the Omo-Gibe river distance from Gibe river upstream to Gibe IV. The model does not include the 452 km of the Omo-Gibe river length from Gibe IV downwards to Lake Turkana in Kenya (Figure 2). The basin has high elevation variation ranging from 350 to 3600 (m.a.s.l), as well as a varying river area slope from Gibe I to Gibe IV (>40%) (Figure 3). The high elevation and slope variation along the Omo-Gibe river gives a plausible condition to construct a cascade of hydropower plants along the river to generate hydropower from high river falls. On the other side, there are conventional stable riverside areas in the upstream areas around the Tolay district and downstream after the Gibe_IV hydropower station. These stable areas have a vertical slope ranging from 0–20% (Figure 3). The river's stable areas with 0–20% slope are ideal for medium to large-scale agriculture. The basin produces an annual flow of 16.6 BMC (billion cubic meters) runoff at the Omo river outlet, i.e., 14% of the country`s annual surface water resources [18]. The river basin is characterized by three different agro climatology zone, i.e., cold zone (Dega), temperate zone (Weyna dega), and hot zone (Kolla) [19]. The upstream north and the northwest highland around Jimma and Gojeb have a cold climate (temperate oceanic climate). The central highlands areas have a temperate zone, and the southern areas towards the Omo valley have a hot (warm semi-arid) climate. It has an average annual rainfall between 300 mm in the southern lowland to 2000 mm in the northwestern and central highland areas. The rainfall has unimodal (high peak during June-September) in the northern part and bimodal (first peak in April and second peak in October) in the southern part of the basin [20]. For instance, the average monthly precipitation characteristics for upstream (e.g., Baco_station) and lower stream areas (e.g., Sawula_station) are illustrated in Figure 4. The average annual stream flow of the Gibe river near Abelti town, as measured by the gauging, is approximately 6440 Mm³/year (Figure 5).The average annual temperature varies from 17 °C in the northern highlands and 29 °C in the southern lowland [21]. A large part of the basin is dominated by cultivated, woodland, and pastoral land use. Small-scale agricultural farming activities are commonly practiced in the northern and central parts of the basin. Large-scale agriculture like sugarcane and cotton farms are common in the southern lowland areas.

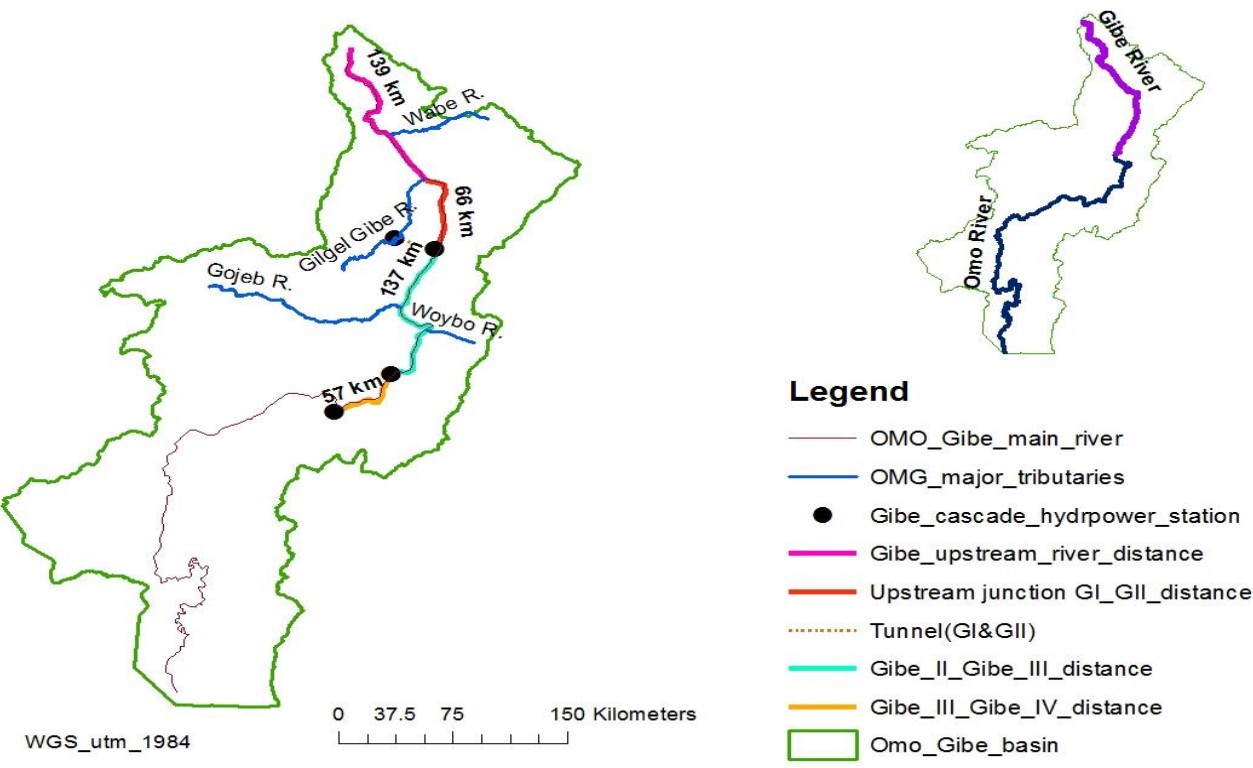

**Figure 2.** Omo-Gibe River length and tributaries considered in the model.

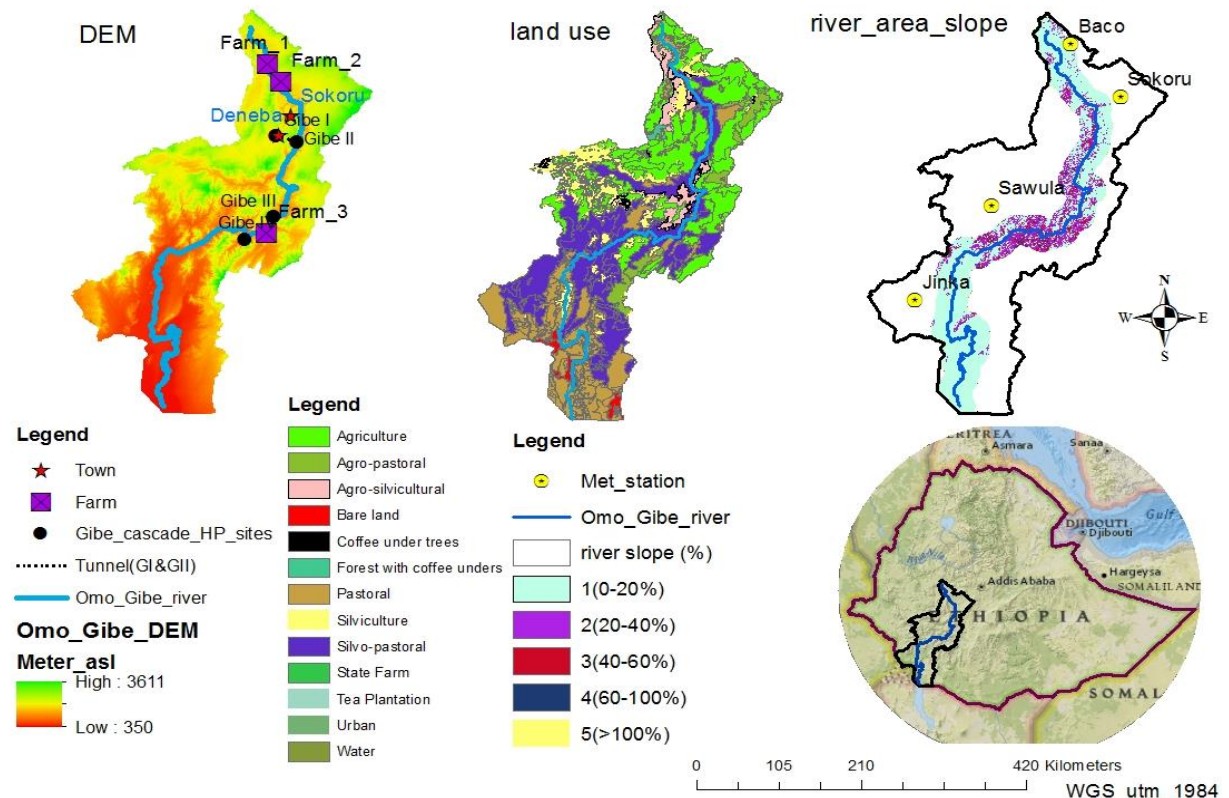

**Figure 3.** Pictures of the Omo Gibe river basin showing the elevation, towns, Gibe cascade hydropower plants, land use distribution, the slope along the river length, and agricultural farms (data sources FAO, DAFNE database, 2019).

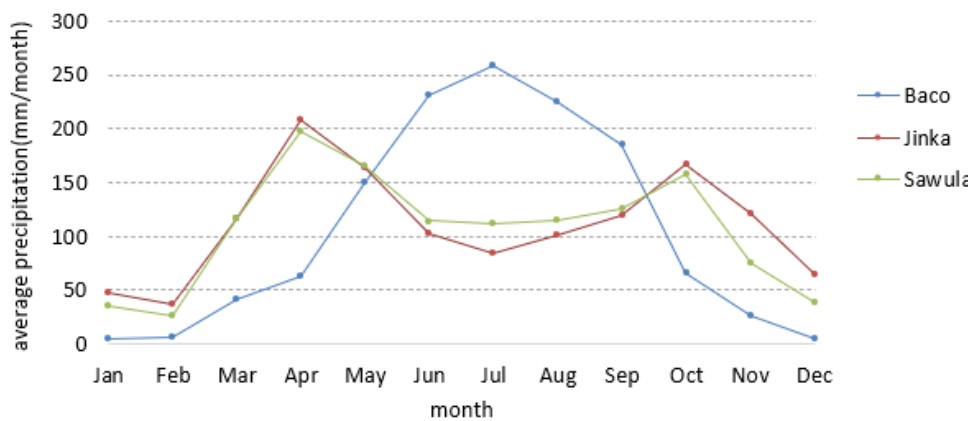

**Figure 4.** Average monthly precipitation for weather stations at Baco (1998–2017), Jinka (1998–2018), and Sawula (1998–2018) [21].

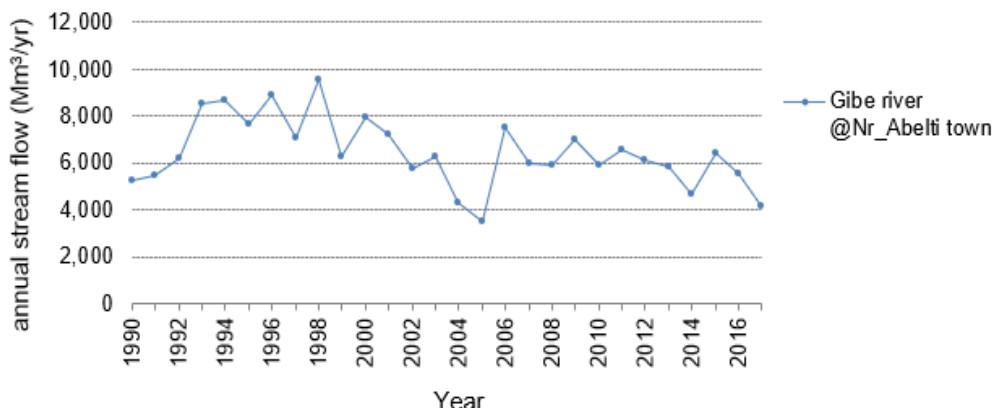

**Figure 5.** Annual stream flow of Gibe river at the gauge station near the Abelti town (1990–2016) [22].

*2.2. Materials*

In this study, different agricultural land-based environmental and economic attributes data were collected and used as model input, such as crop production cost, crop income, food calorie production, soil loss, nitrate leaching, and soil organic carbon sequestration per unit season per land unit. Likewise, water-based data were also used, such as crop water requirement, rainfall, streamflow, water demand for hydropower to generate energy, and water supply for domestic city requirements [21,22]. The data were arranged and organized on a weekly basis for a whole year per hectare of land. Some of the data were obtained from field observations and laboratory analyses. The rest of the data were collected from secondary sources. Crop production yield was based on field data [23]. Food calorie production content and authors' own synthesis were based on [24]. Nitrate leaching data was generated based on field visits and authors' own calculations based on [23,25,26]. Soil loss data was based on [27,28]. Crop income and crop production cost were based on a field visit and authors' own calculations based on [23,29]. Crop growing length and crop water requirement data for different crops and vegetables were obtained from Araya et al. [30], Brouwer et al. [31], and Ten Berge et al. [32].

Alternative crop successions for the Omo-Gibe basin

It is a common agricultural practice to plant a particular crop on a given piece of land during a season. Some farmers choose to harvest once per year, while others may harvest multiple times, depending on various factors such as the availability of agricultural

inputs, water sources like rain or irrigation, budget constraints, soil fertility, land availability, etc. For example, a farmer may decide to plant a particular crop on their landing in a given season, or they may choose to leave the land as fallow. The alternative crop successions format is designed for different combinations of crops and vegetables, taking into account the length of their growing season (Figure 6). The Gebre-model optimizes the allocation of a specific land unit to a specific alternative crop succession based on the objectives and constraints functions. In this study, 12 crop plants and vegetables, namely barley (B), green beans (Bn), cabbage (C), carrot (Cr), dry onion (o), pepper (p), potato (Po), maize (M), sorghum(S), teff (T), tomato (To), and wheat (W) that are commonly used by farmers in the study area were selected to formulate 20 alternative crop succession as alternative land use options (Figure 6). In addition, the fallow land use option was also considered in the alternative crop successions (F). The alternative crop succession covers a 12-month period starting in March. March was selected as the initial sowing month because it marks the beginning of the Belg (spring) season in the upper and lower stream areas. In this study, three farms and four 300-ha land units were considered. Each land unit has varying agricultural performance due to differences in soil, landscape, and input use.

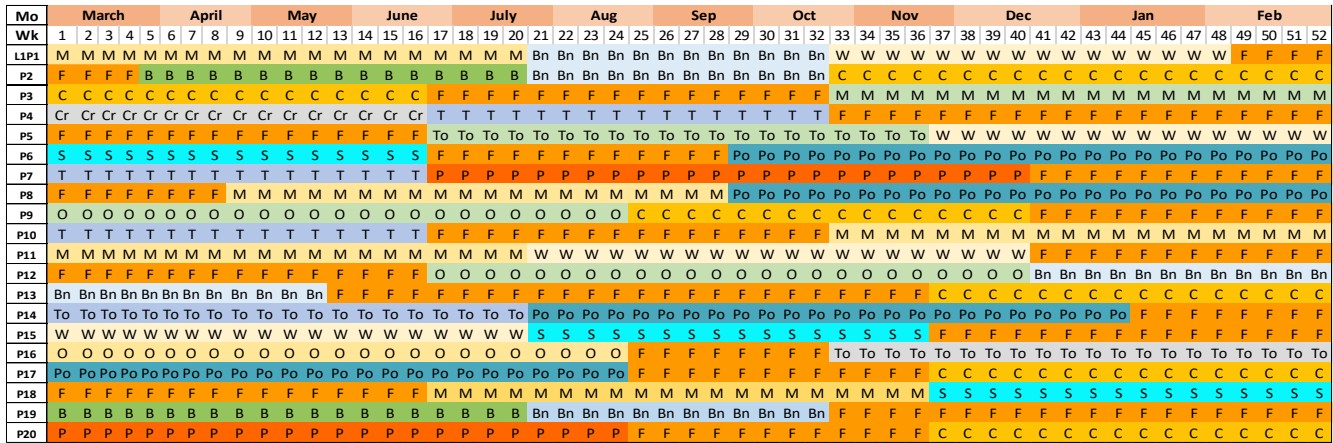

**Figure 6.** Alternative crop successions format used in the model (M: maize; Bn: bean; W: wheat; F: fallow; B: barley; C: cabbage; Cr: carrot; T: teff; To: tomato; S: sorghum; Po: potato; O: onion; P: pepper), Mo: month; WK: week; L1P1: alternative crop succession to land unit 1.

Different input parameters were prepared to fit into the generated alternative crop succession proposal format. The input parameters were food calorie ($cal_{pjft}$), crop income ($ci_{pjft}$), crop production cost ($cc_{pjft}$), crop water requirement ($cwr_{pjft}$), nitrate leaching ($nl_{pjft}$), soil loss ($sl_{pjft}$), and soil organic carbon sequestration ($soc_{pjft}$) (Figure 7). The input parameter data for each land unit per week were designed according to the input parameter characteristics. For example, food calories and crop income were expected only at the end of the growing season (harvesting week) (Figure 8a,b). The input parameter values were set to zero for the remaining growing weeks. The other input parameters had different characteristics in space and time. Crop production cost, nitrate leaching, and crop water requirement per land unit per season input parameters had a relatively normal distribution graph pattern (Figure 8c,d,g). Whereas soil loss had a declining pattern (Figure 8e). During the early stages of the growing period, when crop coverage is low, soil loss is significant. However, as the crop matures and grows, the soil loss decreases, and the soil organic carbon sequestration value increases (as depicted in Figure 8f). The input parameter values in the model were designed to reflect this phenomenon.

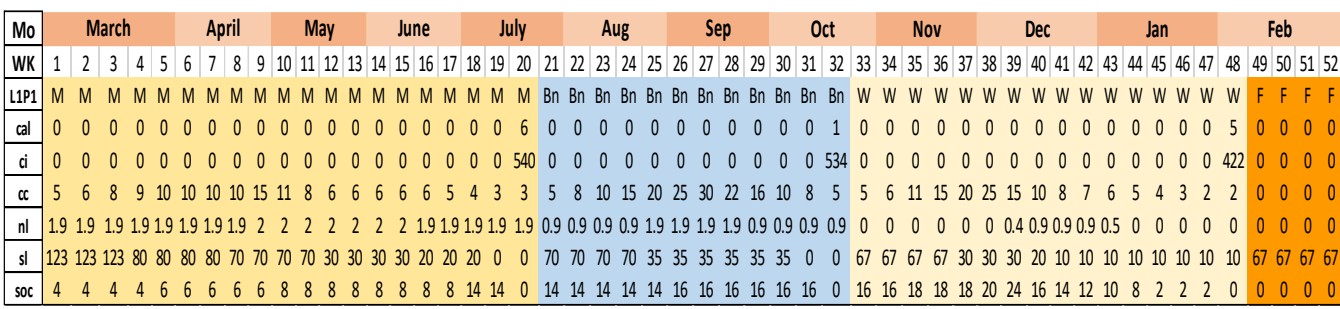

**Figure 7.** Input data used for the land attribute data sets for alternative crop succession type P1 per unit land per week for one year. *(L1P1 land unit 1 for alternative crop succession P1, food calorie(cal in MegKcal/ha/w), crop income (ci in USD/ha/week), crop production cost (cc in USD/ha/week), nitrate leaching (nl in kg /ha/week), soil loss (sl in Kg/ha/week), soil carbon sequestration (soc in Kg/ha/week). For nl, sl, and soc, here expressed in Kg but throughout the model it is used in tons.*

**Figure 8.** Input parameter characteristics used in the model for the alternative crop succession: (**a**) food calorie (ca), (**b**) crop income(ci), (**c**) crop cost (cc), (**d**) nitrate leaching (nl), (**e**) soil loss (sl), (**f**) soil carbon sequestration (soc), (**g**) crop water requirement (cwr).

### 2.3. Methods

The **Gebre mathematical land and water allocation optimization model** was used to simulate the Omo-Gibe river basin land and water allocation to meet water, food, energy, and environmental conservation under climate services through climate-smart land use planning. This model is a linear programming optimization model which has multiple objectives and constraint functions (Table 1). It is conceived to optimize the allocation of

alternative crop successions to land units taking the dynamic availability of water from a river with a reservoir system. It aims to maximize aggregate gross income from crop production in irrigation-based farms and to minimize the associated crop cost, reservoir water travel cost (it is a cost incurred when water travels between the reservoir and river link), and penalties attributed to unmet requirements related to food calorie production, soil organic carbon sequestration, reservoir volume, and nitrate leaching and soil loss through erosion. The model takes into account multiple constraints related to land and water usage. It aims to balance competing demands by minimizing the trade-off and maximizing synergies in order to sustainably allocate limited land and water resources through climate-smart land use allocation. The model determines the optimal allocation of alternative crop successions to land units as well as the optimal allocation of water among competing water use activities.

**Table 1.** Objective and constraint function used in the model.

| Description | Equation | Unit | Eq_No. |
|---|---|---|---|
| Objectives | $$MaxZ = \left[\sum_{p}^{p}\sum_{j}^{J}\sum_{f}^{F}\sum_{t}^{T}(ci_{pjft} * X_{pjf})\right] - \left[\sum_{p}^{P}\sum_{j}^{J}\sum_{f}^{F}\sum_{t}^{T}(cc_{pjft} * X_{pjf})\right]$$ $$-\left[Pcal * \sum_{f}^{F} SCAL_f\right] - \left[Pssoc * \sum_{f}^{F} SSOC_f\right]$$ $$-\left[Penl * \sum_{f}^{F} ENL_f\right] - \left[Pesl * \sum_{f}^{F} ESL_f\right]$$ $$-\left[\sum_{r}^{R} PVR_r * SVR_r\right] - \left[\sum_{d}^{D}\sum_{t}^{T} Pd_d * SYD_{dt}\right]$$ $$-\left[\sum_{r}^{R}\sum_{t}^{T} Pr_r * (YRN_{rt} * YNR_{rt})\right]$$ | USD | (1) |
| Constraints | | | |
| land-based constraints | | | |
| food calorie production | $$\sum_{p}^{P}\sum_{j}^{J}\sum_{t}^{T}(cal_{pjft} * X_{pjf}) + SCAL_f - ECAL_f = Lcal_f$$ | MegKcal | (2) |
| minimum soil organic carbon sequestration demand | $$\sum_{p}^{P}\sum_{j}^{J}\sum_{t}^{T}(soc_{pjft} * X_{pjf}) + SSOC_f \geq Lsoc_f$$ | ton | (3) |
| maximum nitrate leaching | $$\sum_{p}^{P}\sum_{j}^{J}\sum_{t}^{T}(nl_{pjft} * X_{pjf}) - ENL_f \leq unl_f$$ | ton | (4) |
| maximum soil loss | $$\sum_{p}^{P}\sum_{j}^{J}\sum_{t}^{T}(sl_{pjft} * X_{pjf}) - ESL_f \leq usl_f$$ | ton | (5) |
| maximum available land area size | $$\sum_{p}^{P} X_{pjf} \leq A_{jf}$$ | ha | (6) |
| minimum farm income expectation limit | $$\sum_{p}^{P}\sum_{j}^{J}\sum_{t}^{T}(ci_{pjft} * X_{pjf}) + SCI_f \geq Lci_f$$ | USD | (7) |
| maximum crop production budget | $$\sum_{p}^{P}\sum_{j}^{J}\sum_{t}^{T}(cc_{pjft} * X_{pjf}) \leq ucc_f$$ | ton | (8) |
| constraint(combined land and water) | | | |
| crop water requirement demand | $$\sum_{p}^{P}\sum_{j}^{J}(cwr_{pjft} * X_{pjft}) = FCWR_{ft}$$ | m³/t | (9) |

| | | | |
|---|---|---|---|
| rainfall requirement or available | $\sum_{p}^{P}\sum_{j}^{J}\left(rw_{pjft} * X_{pjft}\right) = FRW_{ft}$ | m³/t | (10) |
| irrigation water requirement | $FRW_{ft} + IRR_{ft} = FCWR_{ft}$ | m³ | (11) |
| water based constraint | | | |
| initial condition | | | |
| | river segment(n) | | |
| | $Y_n^0 = Y_{n+1}^{initial}$ | m³/t | (12) |
| | $VN_n^0 = VN_{n+1}^{initial}$ | m³ | (13) |
| | reservoir segment(r) | | |
| | $VR_r^0 = VR_r^{initial}$ | m³ | (14) |
| | transport(n) | | |
| | river segment(n) | | |
| river link flow continuity constraint based on the muskingum method | $Y_{n+1,t+1} = C_n^0 * Y_{n,t+1} + C_{n,t}^1 * Y_{n,t} + C_n^2 * Y_{n+1,t}$ | m³/t | (15) |
| | $C^0 := \dfrac{-KW + 0.5\,\Delta t}{K - KW + 0.5\Delta}$ | - | (16) |
| | $C^1 := \dfrac{KW + 0.5\,\Delta t}{K - KW + 0.5\Delta}$ | - | (17) |
| | $C^2 := \dfrac{K - KW - 0.5\,\Delta t}{K - KW + 0.5\Delta}$ | - | (18) |
| | $C^0 + C^1 + C^2 = 1$ | - | (19) |
| flow/transport balance constraint | | | |
| | river segment | | |
| | $VN_{n,t+1} = VN_{n,t} + \left(Y_{n,t} - Y_{n+1,t}\right)$ | m³ | (20a) |
| | $VN_{n,t+1} = VN_{n,t} + \left(Y_{n,t} + YZN_{n,t}\right) - \left(Y_{n+1,t} + YRN_{r,t} - YNR_{r,t} - IRR_{f,t} - YD_{d,t} - Qo_{n,t}\right)$ | m³ | (20b) |
| | reservoir link | | |
| | $VR_{r,t+1} = VR_{r,t} + \left(YNR_{r,t} - YRN_{r,t} - YD_{d,t}\right)$ | m³ | (21a) |
| | $VR_{r,t+1} = VR_{r,t} + \left(YNR_{r,t} + YZR_{r,t} - YRN_{r,t} - IRR_{f,t} - YD_{d,t}\right)$ | m³ | (21b) |
| | river segment | | |
| capacity constraints | $Y_{n,t} + SY_{n,t} \geq Ymin_n$ | m³/t | (22) |
| | $Qo_{n,t} \geq Qomin_n$ | m³/t | (23) |
| | $Qo_{n,t} \leq Qomax_n$ | m³/t | (24) |
| | $VN_{n,t} \geq VNmin_n$ | m³ | (25) |
| | $VN_{n,t} \leq VNmax_n$ | m³ | (26) |
| | reservoir segment | | |
| | $VR_{r,t} + SVR_r \geq VRmin_r$ | m³ | (27) |
| | $VR_{r,t} + SVR_r \leq VRmax_r$ | m³ | (28) |
| | $YRN_{r,t} + IRR_{f,t} + YD_{d,t} \leq Resmaxoutflow_r$ | m³/t | (29) |
| | demand link | | |
| | $YD_{d,t} + SYD_{d,t} \geq Dmin_d$ | m³/t | (30) |
| | $YD_{d,t} \leq Dmax_d$ | m³/t | (31) |

The mathematical model was coded and optimized using the LINGO programming language [33]. It is a linear interactive general optimization (LINGO) mathematical programming language used to solve complex linear, nonlinear, and integer optimization problems. It was used in this study because of its flexibility and interactiveness. Moreover, it is useful to solve complex, large-scale problems [34]. This model combines land and water-based functions to provide a comprehensive solution to meet the demands for water, food, energy, environmental conservation and river ecosystem preservation, under the concept of a climate-smart land use planning system for the Omo-Gibe river basin.

2.3.1. Model Set Up

The model has three irrigation-based farm sites located along the Omo-Gibe river. The first two are situated in the upper stream of the river around the Tolay district. The third farm is located after the Gibe_III along the periphery of the Gofa zone in the Sawula district. These farms are medium-scale irrigation-based agricultural areas where farmers divert water and use it for irrigation purposes. Two close towns are considered one-demand nodes (Deneba and Sokoru) with 25,000 inhabitants [35]. These towns are closely linked to the Omo-Gibe river and extract water from the river for municipal use. Furthermore, there are four cascade hydropower stations. At Gibe_I (184MW), water is subtracted from the river to the reservoir and then diverted from the dam to the underground tunnel to the power station. The water after power generation passes to a long underground tunnel (26 km) to Gibe_II power station to generate hydropower power (420 MW) [17]. After hydropower generation, the water is discharged to the river channel. The Gibe_I reservoir dam is a rock-filled embankment dam. It has 1700 m in length and 40-m height with a 917 million cubic meters reservoir capacity. Gibe III (1870 MW) is a 243-m high, and 610 m crest length roller compacted concert dam. It has a total reservoir capacity of 14,700 million cubic meters [36]. The Gibe_IV dam is under construction and is expected to generate 1472 MW of hydropower energy with an average reservoir volume of 10 BCM [17]. According to the Ethiopia Electric Corporation, cascade hydropower generation power plants play a great role in securing the growing energy demand across the country and in the larger East African region. Hence, in this study, around 400 km stretch of the river distance was considered, from the northern tip of the river (i.e., near Baco) to the beginning of the southern Omo lowlands (Figure 9).

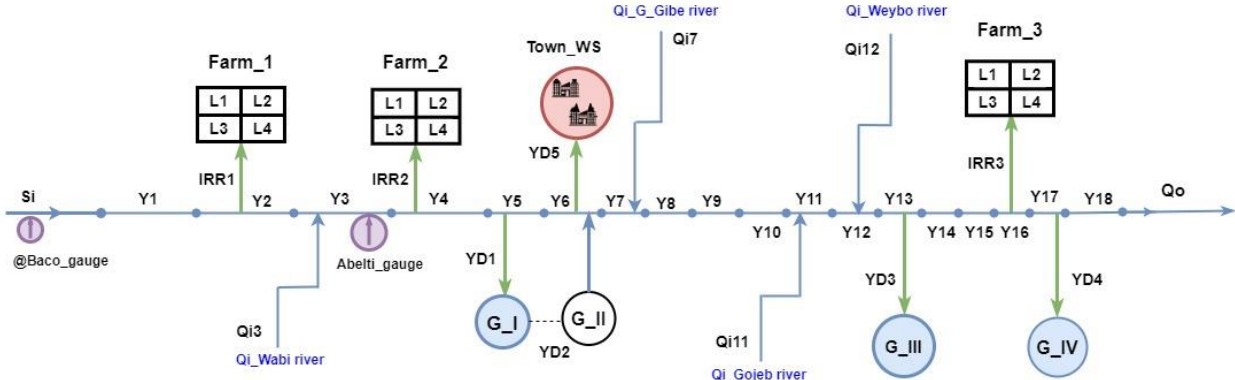

**Figure 9.** Diagram of the land and river network (it has 3 farms (F1(IRR1), F2(IRR2), F3(IRR3), 4 land units (L1…L4), and 18 river segments (Y1…Y18), 4 hydropower stations/demand nodes (YD1…YD4) and 1 water supply demand node (YD5). The gauge stations are near Baco town (supply inflow) and Abelti (around the main bridge on the main road to Jimma are indicated on the diagram.

Objective functions and constraints

The objective function is presented in Equation (1). Equations (2)–(8) are specifically for the land-attributed constraints. Equations (9)–(11) are for combined land and water-based constraints, and Equations (12)–(31) are for water-based constraints. The model's initial condition for river/reservoir flow and volume conditions are shown in equations (12)–(14); transport constraints are illustrated in Equations (15)–(21); and capacity constraints for the river, reservoir, and demand links are indicated from Equations (22)–(31).The indices, parameters and variables used in the model equations are presented in Table 2.

**Table 2.** Indices, parameters, and variables used in the model.

| Type | Notation | Description | Unit |
|---|---|---|---|
| indices | d | demand node d ∈ D | - |
| | f | farm f ∈ F | - |
| | j | land unit j ∈ J | - |
| | n | river link n ∈ N | - |
| | P | alternative crop succession p ∈ P | - |
| | r | reservoir link r ∈ R | - |
| | t | period t ∈ T | - |
| parameters | | | |
| | $A_{jf}$ | area of land unit (j) per farm(f) | ha |
| | $cal_{pjft}$ | food calorie per alternative crop successions (p)per land unit(j)per farm(f)per period (t) | MegKcal/ha/t |
| | $cc_{pjft}$ | crop production cost of alternative crop successions (p) per land unit(j)per farm(f)per period (t) | USD/ha/t |
| | $ci_{pjft}$ | crop income per alternative crop successions (p)per land unit(j)per farm(f)per period (t) | USD/ha/t |
| | $C^o, C^1, C^2$ | routing coefficient | - |
| | $cwr_{pjft}$ | crop water requirement per alternative crop successions (p) per land unit(j)per farm(f)per period (t) | m³/ha/t |
| | $Dmin_{dt}$ | demand volume water needed at node d per period(t) | m³/t |
| | $Dmax_{dt}$ | maximum demand volume water needed at node d per period(t) | m³/t |
| | $Lcal_f$ | minimum calorie requirement per farm,∀(f) ∈f | USD |
| | $Lci_f$ | minimum crop income requirement per farm,∀(f) ∈f | USD |
| | $Lsoc_f$ | minimum soil organic carbon sequestration demand per farm(f), ∀(f) ∈f | ton |
| | $nl_{pjft}$ | nitrate leaching of alternative crop successions (p) per land unit(j)per farm(f)per period (t) | ton/ha/t |
| | Pcal | penalty for unmet food calorie production at farm | USD/MegKcal |
| | Pd | penalty for unmet demand at demand link d | USD/m³ |
| | Pr | cost for water travel to and from reservoir to river link r. | USD/m³ |
| | pvr | penalty for unmet reservoir volume of minimum requirement at reservoir r | USD/m³ |
| | $Qomax_{nt}$ | maximum outflow at downstream end river link(n) | m3/t |
| | $Qomin_{nt}$ | minimum outflow demand at downstream river link(n) | m³/t |
| | $Si_{n,t}$ | upstream inflow at the start of river link(n) per period(t) | m³/t |
| | $Resmaxoutflow_{rt}$ | reservoir outflow maximum limit t; ∀(r) ∈ r | m³/t |

| | | | |
|---|---|---|---|
| | $rw_{ft}$ | rainwater per crop pattern (p) per farm(f) per period(t) | $m^3/ha/t$ |
| | $sl_{pjft}$ | soil loss of alternative crop successions (p) per land unit(j)per farm(f)per period (t) | $ton/ha/t$ |
| | $SOC_{pjft}$ | soil organic carbon sequestration of alternative crop successions (p) per land unit(j)per farm(f)per period (t) | $ton/ha/t$ |
| | $Ucc_f$ | maximum crop production budget allowed per farm,$\forall(f) \in f$ | USD |
| | $unl_f$ | maximum nitrate leaching demand per farm(f),$\forall(f) \in f$ | ton |
| | $usl_f$ | maximum soil loss demand per farm(f), $\forall(f) \in f$ | ton |
| | $VNmax_n$ | maximum water volume needed on river link n), $\forall(n) \in n$ | $m^3$ |
| | $VNmin_n$ | minimum water volume needed on river link n), $\forall(n) \in n$ | $m^3$ |
| | $VN^{t=0}$ | initial water volume on river link n, $\forall(n) \in n$ | $m^3$ |
| | $VRinitialr^{t=0}$ | initial reservoir volume on reservoir r,$\forall(r) \in r$ | $m^3$ |
| | $VRmax_r$ | maximum reservoir volume on reservoir link r, $\forall(r) \in r$ | $m^3$ |
| | $VRmin_r$ | minimum of reservoir volume limit, $\forall(r) \in r$ | $m^3$ |
| | $w$ | the Muskingum weighting factor of river link (n,); $\forall(n) \in n$ | - |
| | $Ymin_n$ | minimum water flow needed on river link (n), $\forall(n) \in n$ | $m^3/t$ |
| | $Y_n^{t=0}$ | initial water inflow on river link n, $\forall(n) \in n$ | $m^3/t$ |
| variables | | | |
| | $ECAL_f$ | excess calorie production per farm f, $\forall(f) \in f$ | MegKcal |
| | $ENL_f$ | excess nitrate leaching per farm f, $\forall(f) \in f$ | ton |
| | $ESL_f$ | excess soil loss per farm f, $\forall(f) \in f$ | ton |
| | $FCWR_{ft}$ | total crop water requirement for allocated crop per farm per time t; $\forall(f) \in f$ | $m^3/t$ |
| | $FRW_{ft}$ | total rainwater available for allocated alternative crop successions per farm per time t; $\forall(f) \in f$ | $m^3/t$ |
| | $IRR_{ft}$ | irrigation water allocated for allocated alternative crop successions per farm f per period time t; $\forall(f) \in f$ | $m^3/t$ |
| | $Qo_{nt}$ | downstream river link end outflow at river link n at time t; $\forall(n) \in n$ | $m^3/t$ |
| | $SCAL_f$ | unmet food calorie production per farm f, $\forall(f) \in f$ | MegKcal |
| | $SSOC_f$ | unmet soil organic carbon sequestration per farm | ton |
| | $SYD_{dt}$ | unmet demand at demand link d at time t | $m^3/t$ |
| | $SVR_r$ | unmet reservoir minimum volume capacity limit of reservoir r ,$\forall(r) \in r$ | $m^3$ |
| | $VN_{,n}$ | water volume on river link n at beginning of time t ;$\forall(n) \in n$ | $m^3$ |
| | $VR_r$ | reservoir volume r ,$\forall(r) \in r$ | $m^3$ |
| | $X_{pjf}$ | allocated cropland per alternative crop successions p,per land unit j,per farm f, $\forall(p) \in p, \forall(j) \in j, \forall(f) \in f$ | ha |

| | | |
|---|---|---|
| $YD_{dt}$ | water allocation on-demand link or demand node d at time t; $\forall(d) \in d$ | m³/t |
| $YNR_{rt}$ | reservoir inflow from river link to reservoir link r, at time t $\forall(r) \in r$ | m³/t |
| $Y_{nt}$ | water flow on river link n at time t $\forall(n) \in n$ | m³/t |
| $YRN_{rt}$ | reservoir outflow from river link to reservoir link r, at time t $\forall(r) \in r$ | m³/t |

2.3.2. Input Data Used in the Model

Precipitation
Effective precipitation
Effective precipitation is the amount of rainfall or precipitation left after a runoff, evaporation, and deep percolation. Only the water retained in the root zone can be used by the plants, and it is called the effective part of rainwater. Indirectly, it is the amount of rainwater that is useful for crop growth [37]. It can be calculated as precipitation minus standard evapotranspiration (ETc). The average precipitation for each farm near the weather station (i.e., Farm 1_Baco, Farm 2_Sokoru, and Farm 3_Sawula) is shown in (Table 3).

**Table 3.** Average precipitation in the Omo_Gibe river basin (mm/month).

| Year | | Jan | Feb | Mar | Apr | May | Jun | Jul | Aug | Sep | Oct | Nov | Dec |
|---|---|---|---|---|---|---|---|---|---|---|---|---|---|
| 1998_2017 | Baco | 4.7 | 5.8 | 42.0 | 63.6 | 150.7 | 231.5 | 258.3 | 225.3 | 184.6 | 66.1 | 26.5 | 4.4 |
| average eff.p | farm1 | 0 | 0 | 0 | 0 | 9.3 | 106.7 | 141.8 | 121.2 | 91.0 | 0 | 0 | 0 |
| 1987_2014 | Sokoru | 32.7 | 33.6 | 79.0 | 115.9 | 149.7 | 203.1 | 237.1 | 222.1 | 172.7 | 77.2 | 25.8 | 22.4 |
| average eff.p | farm2 | 0 | 0 | 0 | 7.9 | 30.7 | 95.1 | 135.4 | 130.3 | 91.1 | 5.3 | 0 | 0 |
| 1998_2018 | Sawula | 35.4 | 26.9 | 116.6 | 197.5 | 165.5 | 114.1 | 111.7 | 115.2 | 125.4 | 157.7 | 74.7 | 38.6 |
| average eff.p | farm3 | 0 | 0 | 12.4 | 87.1 | 46.5 | 0 | 2.6 | 11 | 34.2 | 78.3 | 9.9 | 0 |

Data source: [21]; average eff.p:average effective precipitation depth (mm/month).

Standard evapotranspiration (ETc) is defined as reference evapotranspiration (ETo) multiplied by a crop coefficient (Kc). In this case, the Kc was estimated based on the FAO report for different crops [37]. The ETo is estimated using the FAO 56 Penman-Monteith equation [37]. Then the ETo calculator application software developed by Raes,[38] was used to calculate the reference evapotranspiration (ETo). It uses the minimum and maximum temperature, geographical location, altitude, wind speed, humidity, sunshine, and radiation as input to estimate the reference evapotranspiration. The maximum and minimum temperature and estimated reference evapotranspiration for each farm area (weather stations) are presented (Figures 10–12).

Additional data  for the annual supply inflow near the Baco town gauge is illustrated to understand the Gibe river flow pattern (Figure 13). Furthermore, the average monthly stream data for each tributary's inflow used in the model is shown in (Table 4).The input data for the land-based constraint is presented in (Table 5),while the river link constraint is presented in Table 6. Table 7 is used for the reservoir link and (Tables 8 and 9) are used for the demand link constraint part.

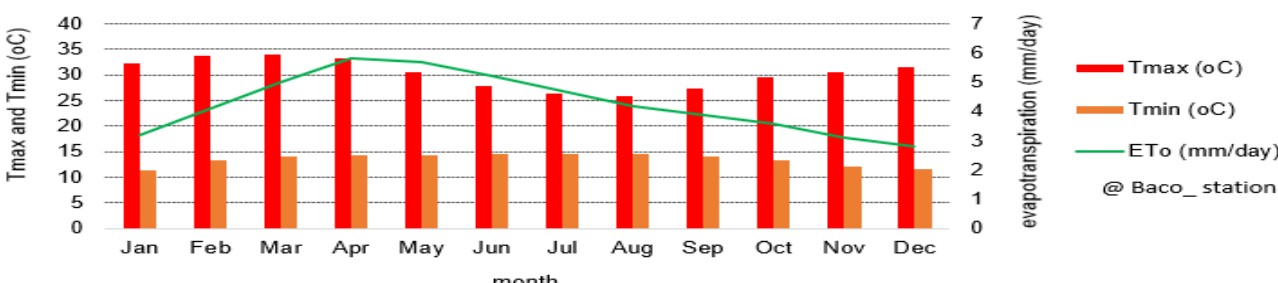

**Figure 10.** Average maximum and minimum temperature and evapotranspiration at Baco weather station (1998–2017).

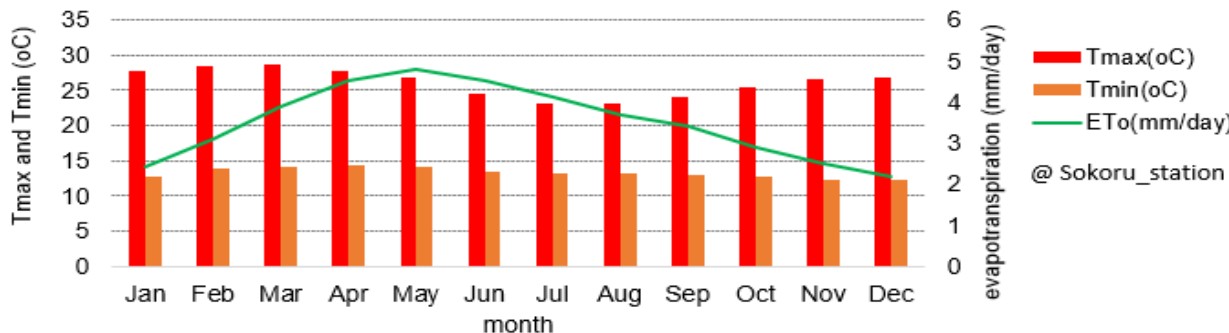

**Figure 11.** Average maximum and minimum temperature and evapotranspiration at Sokoru weather station (1987–2014).

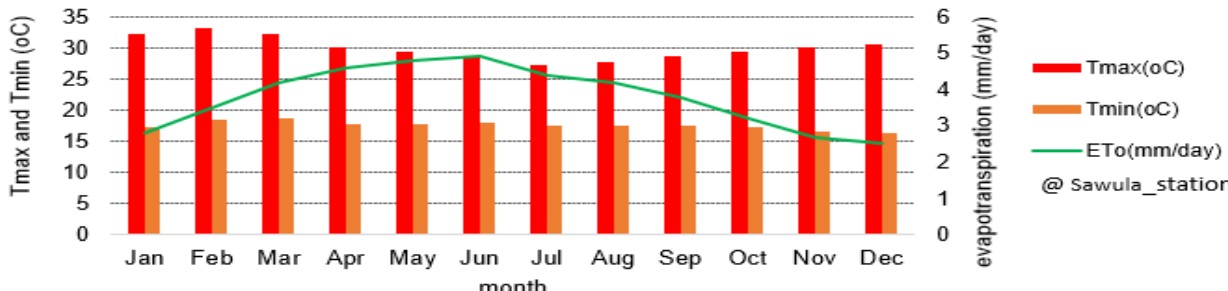

**Figure 12.** Average maximum and minimum temperature and evapotranspiration at Sawula weather station (1998–2018).

Hydrological data

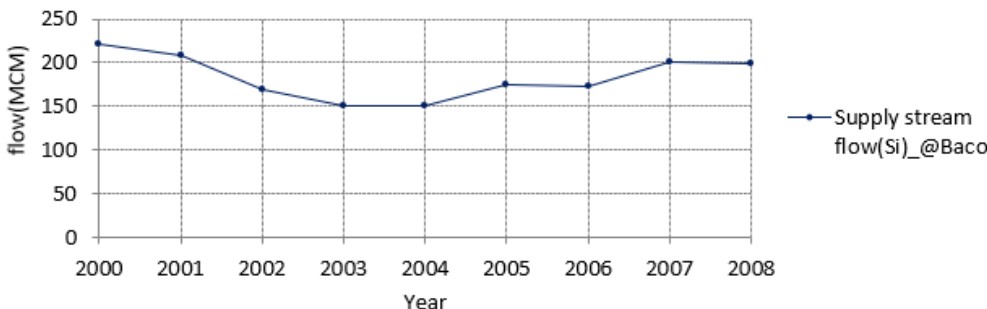

**Figure 13.** Annual supply flow to Gibe river near Baco gauging station.

Table 4. Average monthly river inflow (Mm³/month).

| Year | St | Area | Jan | Feb | Mar | Apr | May | Jun | Jul | Aug | Sep | Oct | Nov | Dec |
|---|---|---|---|---|---|---|---|---|---|---|---|---|---|---|
| 2000–2008 | Supply_Nr_Baco | - | 11.7 | 6.7 | 6.9 | 4.9 | 7.0 | 14.7 | 22.4 | 34.8 | 30.6 | 18.4 | 12.1 | 13.0 |
| 2000_2019 | Gibe_Nr_Asendabo | 2966 sq Km | 22.8 | 16.2 | 18.3 | 26.3 | 67.6 | 113.4 | 241.5 | 310.1 | 284.1 | 123.4 | 58.6 | 31.9 |
| 2000_2017 | Gojeb_nr_Shebe | 3577.0 sq km | 32.5 | 18.9 | 26.1 | 52.4 | 118.2 | 182.6 | 321.8 | 379.2 | 396.6 | 268.9 | 138.1 | 65.6 |
| 1992_2006 | Weybo_nr_Areka | 2368.4 sq km | 0.9 | 0.6 | 1.0 | 2.0 | 3.9 | 3.0 | 7.5 | 12.8 | 7.6 | 9.7 | 5.0 | 2.6 |
| 2000_2007 | Wabi_nr_Welkite | 1866.0 sq km | 9.9 | 11.2 | 16.7 | 36.4 | 32.3 | 78.8 | 255.0 | 340.8 | 150.2 | 46.4 | 13.2 | 8.3 |
| 1990_2017 | Gibe_Nr_Abelti | 15746 sq Km | 69.2 | 94.1 | 94.4 | 108.3 | 120.1 | 328.4 | 1013.7 | 1784.0 | 1360.6 | 935.7 | 341.4 | 190.2 |

Data source [22] Gibe_near_Abelti town is the main river channel.

Table 5. Input data used for land-based constraints.

| Type | (MegKcal/Year) | Lci (USD/Year) | ucc (USD/Year) | Lsoc (ton/Year) | unl (ton/Year) | usl (ton/Year) | A (ha) | Total Farm Size (ha) | Land-Based Penalty | |
|---|---|---|---|---|---|---|---|---|---|---|
| Farm1 | 8000 | 1,500,000 | 462,988.8 | 465.3 | 86.4 | 3815.5 | 300 | 1200 | penalty for unmet food calorie production(USD/MegKcal/year)_Pcal | 9 |
| Farm2 | 10,000 | 1,500,000 | 483,776.9 | 491.7 | 82.7 | 3261.8 | 300 | 1200 | penalty for unmet soil organic carbon sequestration (USD/ton/year)_Pssoc | 5 |
| Farm3 | 7000 | 1,500,000 | 388,814.7 | 449.5 | 42 | 2837.6 | 300 | 1200 | penalty for excess nitrate leaching(USD/ton/year)_Penl | 6 |
| | | | | | | | | | penalty for excess soil loss(USD/ton/year)_Pesl | 8 |

Lcal (minimum food calorie production demand per farm); Lci (minimum crop income requirement per farm); ucc (maximum allowable crop production budget per farm); Lsoc (minimum soil organic carbon sequestration); Unl (maximum nitrate leaching limit); Lsl (maximum soil loss limit); A (maximum available land unit size in ha per year).

**Table 6.** Input data used in the model for river link part constraints.

| Link | Node | | | | Routing Coefficient | | | River Link Flow (Mm³) | | River Link Volume Capacity (Mm³) | | | Minimum/Maximum River End Outflow (Mm³) | |
|---|---|---|---|---|---|---|---|---|---|---|---|---|---|---|
| River Link (n) | from | to | k | w | C0 | CONE | CTWO | Ymin | Yinitial | VNmin | VNmax | VNinitial | Qomin | Qomax |
| 1 | 1 | 2 | 1 | 0.25 | 0.2 | 0.6 | 0.2 | 0.5 | 0 | 95 | 1100 | 500 | - | - |
| 2 | 2 | 3 | 1 | 0.25 | 0.2 | 0.6 | 0.2 | 1 | 1.8 | 100 | 1200 | 500 | - | - |
| 3 | 3 | 4 | 1 | 0.25 | 0.2 | 0.6 | 0.2 | 1 | 1.8 | 100 | 1200 | 500 | - | - |
| 4 | 4 | 5 | 1 | 0.25 | 0.2 | 0.6 | 0.2 | 1 | 1.8 | 100 | 1200 | 500 | - | - |
| 5 | 5 | 6 | 1 | 0.25 | 0.2 | 0.6 | 0.2 | 1 | 1.8 | 100 | 1200 | 500 | - | - |
| 6 | 6 | 7 | 1 | 0.25 | 0.2 | 0.6 | 0.2 | 1 | 2.5 | 100 | 3629.9 | 500 | - | - |
| 7 | 7 | 8 | 1 | 0.25 | 0.2 | 0.6 | 0.2 | 1 | 2.5 | 100 | 11,094.2 | 500 | - | - |
| 8 | 8 | 9 | 1 | 0.25 | 0.2 | 0.6 | 0.2 | 1.1 | 3 | 100 | 8378.13 | 500 | - | - |
| 9 | 9 | 10 | 1 | 0.25 | 0.2 | 0.6 | 0.2 | 1.2 | 3 | 100 | 8178.5 | 500 | - | - |
| 10 | 10 | 11 | 1 | 0.25 | 0.2 | 0.6 | 0.2 | 1.2 | 3 | 100 | 7282.5 | 500 | - | - |
| 11 | 11 | 12 | 1 | 0.25 | 0.2 | 0.6 | 0.2 | 2 | 4 | 100 | 6622.7 | 500 | - | - |
| 12 | 12 | 13 | 1 | 0.25 | 0.2 | 0.6 | 0.2 | 2 | 4 | 150 | 6100 | 500 | - | - |
| 13 | 13 | 14 | 1 | 0.25 | 0.2 | 0.6 | 0.2 | 2 | 4 | 160 | 17,961.9 | 500 | - | - |
| 14 | 14 | 15 | 1 | 0.25 | 0.2 | 0.6 | 0.2 | 2 | 4.5 | 160 | 6600 | 500 | - | - |
| 15 | 15 | 16 | 1 | 0.25 | 0.2 | 0.6 | 0.2 | 2 | 4.5 | 160 | 6600 | 500 | - | - |
| 16 | 16 | 17 | 1 | 0.25 | 0.2 | 0.6 | 0.2 | 2 | 4.5 | 150 | 5684.1 | 500 | - | - |
| 17 | 17 | 18 | 1 | 0.25 | 0.2 | 0.6 | 0.2 | 2 | 3 | 150 | 5200.6 | 500 | - | - |
| 18 | 18 | 19 | 1 | 0.25 | 0.2 | 0.6 | 0.2 | 2 | 3 | 120 | 5100 | 500 | - | - |
| 19 | 19 | - | | | | | | | | | | | 8.5 | 34 |

**Table 7.** Input data used in the model for reservoir link part constraints.

| Res_Link (n) | Volume (Mm³) | | | Flow (Mm³/Week) | Cost for Water Travel between Reservoir-River Link (USD/Mm³/Week) | Penalty for Unmet Reservoir Storage (USD/Mm³/Week) |
|---|---|---|---|---|---|---|
| | VRinitial | VRmin | VRmax | Resmaxoutflow | pr | Pvr |
| 1 | 917 | 500 | 1300 | 1000 | 0.0003 | 0.5 |
| 2 | 11,750 | 8000 | 16,000 | 1500 | 0.0003 | 0.5 |
| 3 | 10,000 | 7000 | 16,000 | 2000 | 0.003 | 0.5 |

**Table 8.** Input data for the demand node link.

| Demand Type | Demand Node | Minimum Demand Capacity Limit (mm³) | Penalty (USD/Mm³/Week) |
|---|---|---|---|
| | d | Dmin | Pd |
| WS_Sokoru & Deneba | d5 | 0.35 | 7 |

the minimum demand was calculated based on the minimum water required per capita per day (15 Lt) (Ethiopia Ministry of Water and Energy strategy report, 2020).

**Table 9.** Gibe cascade hydropower demands [39–41].

| Gibe Cascade Hydropower Dams | Demand Node | Hydropower Generation Water Demands (Mm³/Week) | Penalty for Unmet Demand (USD/Mm³/Week) |
|---|---|---|---|
| | d | minimum demand | Pd |
| Gibe_I | d1 | 140 | 2 |
| Gibe_II | d2 | 140 (gets water directly from Gibe I) | 2 |
| Gibe_III | d3 | 237.3 | 3 |
| Gibe_IV | d4 | 373.8 | 3 |

## 3. Results

### 3.1. Land-Water-Food-Energy-Environment-Nexus(LWFEEN)

The interlinkage between land resources, water systems, food production, energy resources, environmental conservation, and climate has drawn global attention in the last two decades [42]. In the literature, such kind of interaction is commonly called a water energy-food (WFE) nexus [43]. The water-food-energy nexus decision support system often ignores land and climate aspects, neglecting or omitting some important components with significant impact on resource allocation. It fails to fully address the complex interactions that exist in the nexus approach [44,45]. Land attributes and climate mitigation are essential aspects that should be included in the nexus approach. Land for food (crop production) and food for land (crop production practice activities). Water for food (irrigation and leaching for crop production). Food for water (nourishment for water activities, food resources wastes for water treatment). There is a conceptual practice that shows different types of food resource reactors has a positive impact on wetland treatments. For example, bacteria in food waste have played a significant role in removing nitrate from effluent water [46]. Furthermore, rice cultivation removes phosphorous from drainage water [47]. Water for energy (cooling, hydropower generation) and energy for water (water treatment, transport). Land use for climate change mitigation (crop cultivation for soil carbon sequestration) and climate change mitigation for land use (food crop and soil carbon sequestration crop competition), and so on. Numerous factors drive the interactions: demand increases for food, water, and energy, and concern about environmental degradation such as pollution, land use change, and climate change. There is an urgent need to respond to these complex interactions with a viable decision-support tool through multiple sectoral policies and strategies.

This study recognized the limitations of other models used in nexus research. Therefore, it acknowledged the synergies, conflicts, and trade-offs between land, water, food, energy, environmental conservation, and climate change mitigation interactions. It will aid in providing a comprehensive representation of a full LWFEE nexus under climate change mitigation/river ecosystem services. It will better assess the benefits of land and water allocation for decision-makers. This study focused on the interaction between land-water-food-energy and environment conservation in the notion of climate change mitigation/river ecosystem services in the Omo Gibe river basin. It recognized the complexity of the multiple interactions and sought to address the challenges of balancing natural and socio-economic demands for sustainable development goals. The Omo-Gib river basin is

one of the largest river basins found in Ethiopia. There are multiple water resource development structures developed, and some are under construction along the main river. Additionally, it is a transboundary river flowing into Lake Turkana in Kenya. As a result, there is high competition for land and water resources to meet the growing demands for water, food, energy, ecosystem, and climate change mitigation. This study employed a previously developed Gebre-model to allocate land and water resources under the presumption of land-water-food-energy-environment nexus under the climate change mitigation/river ecosystem services approach. The outputs of the model results were categorized under three sections for better discussion and presenting the interlinkage between the different components.

### 3.2. Land-Water-Food Nexus (LWFN)

This section referred to the interactions between land-water-food production. Agricultural crop productivity requires water for crop consumption. A land resource is used for crop production to produce food. So, when there is a limited land resource, it is complex to respond to how much land is needed to produce enough food and how much water is required to produce the food. Therefore, the availability of land is a determining factor in the amount of food production, notwithstanding other variables like productivity costs and other socio-environmental constraints. Irrigation water is needed for the crop to produce food. The allocation of irrigation water is interdependent on the allocated land resources. When the allocated land size increases, irrigation water allocation demands increase. Conversely, the available irrigation water amount influences the allocation of available land resources. This study summarized the allocated land, irrigation water, and food calorie production.

#### 3.2.1. Land Allocation (ha)

Figure 14 shows the allocated alternative crop successions to land units for the three farm sites (F1, F2, and F2). In this result, relatively different alternative crop successions are assigned. Alternative crop successions (P7 and P8) are often allocated compared to the other alternative crop successions. In general, eight different alternative crop successions are selected out of the proposed 20 alternative crop successions. More than 97 percent of available land is allocated to alternative crop successions (Table 10).

**Figure 14.** Allocated alternative crop succession to land unit per farm (ha).

**Table 10.** Portion of allocated land use with respect to the available land per farm (ha).

| Farm | Land Available (ha) | Allocated Land (ha) | Used % |
|------|---------------------|---------------------|--------|
| F1 | 1200 | 1098.7 | 90 |
| F2 | 1200 | 1200 | 100 |
| F3 | 1200 | 1200 | 100 |

### 3.2.2. Food Calorie Production (MegKcal)

There is an obvious interlinkage between land area and food production. More land means there is a potential for more food production. In this regard, this model has optimized food calorie production per total farm level and met the food demand required (Figure 15). The high food calorie production result in farm 2 is due to the high yield and calorific value of the allocated alternative crop successions. In this alternative crop succession generation, each alternative has different input values due to variations in soil types, farm management, etc., among land units in each farm. Thus, it is expected that each alternative crop succession in each land unit will have different input values.

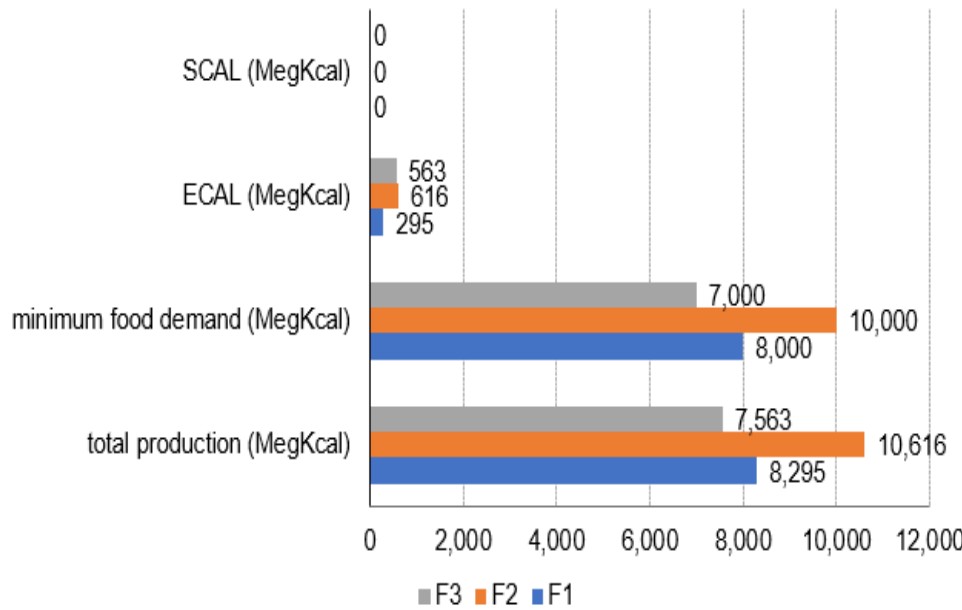

**Figure 15.** Food production produced from the three farms (MegKcal/farm/year).

### 3.2.3. Irrigation Water Allocation

This section presented the dependence between the land-water nexus. Water is needed to meet the crop water requirement to produce food crops or food calories on the land. Normally, the water interacts with land resources, namely for nutrient leaching and drainage activities. However, in this study, the model only considered a one-way interaction of allocated cropland to irrigation water demands.

Figures 16–18 indicate irrigation water allocation and crop water requirement with respect to the allocated cropland for farms 1, 2, and 3. The irrigation water allocation amount is influenced by rainfall availability. Hence, the land allocation is linked to the water availability to meet the crop water requirements. The land allocation size and pattern are highly dependent on irrigation water allocation and available rainfall. Farm 1 and Farm 2 more or less have similar rainfall characteristics. They are both located around the upper Omo-Gibe river basin part, which is linked to river segments 2 and 4, respectively. Farm 3 (Figure 9) is located close to the end of river segment 16. It is just before the Gibe IV hydropower station. This area has a bimodal rainfall pattern that is different from the upper Omo-Gibe part, which has a unimodal pattern.

In summary, this section discussed the linkage between the land-water-food nexus. The allocated land is linked to irrigation water and food production demands. Model results ensure that the food demand is met and excess food is produced. The excess food calorie production can be exported to food-insecure areas. Thus, understanding the interaction of land-water-food supports decision-makers in allocating limited land and water resources to meet the growing food demands.

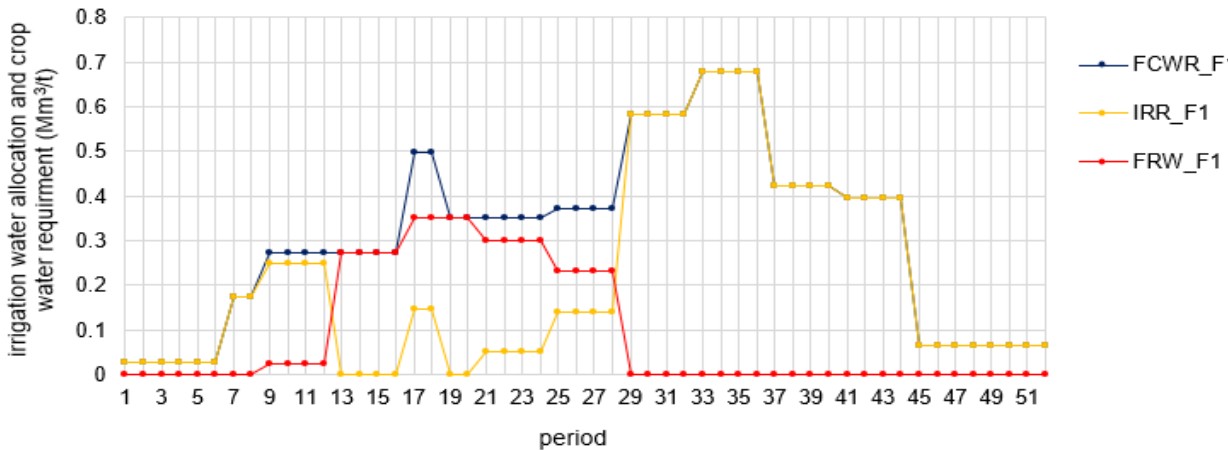

**Figure 16.** Allocated irrigation water (orange), crop water requirement (blue), and available rainfall (red) for farm 1 (Omo-Gibe River basin)(Mm³/week)

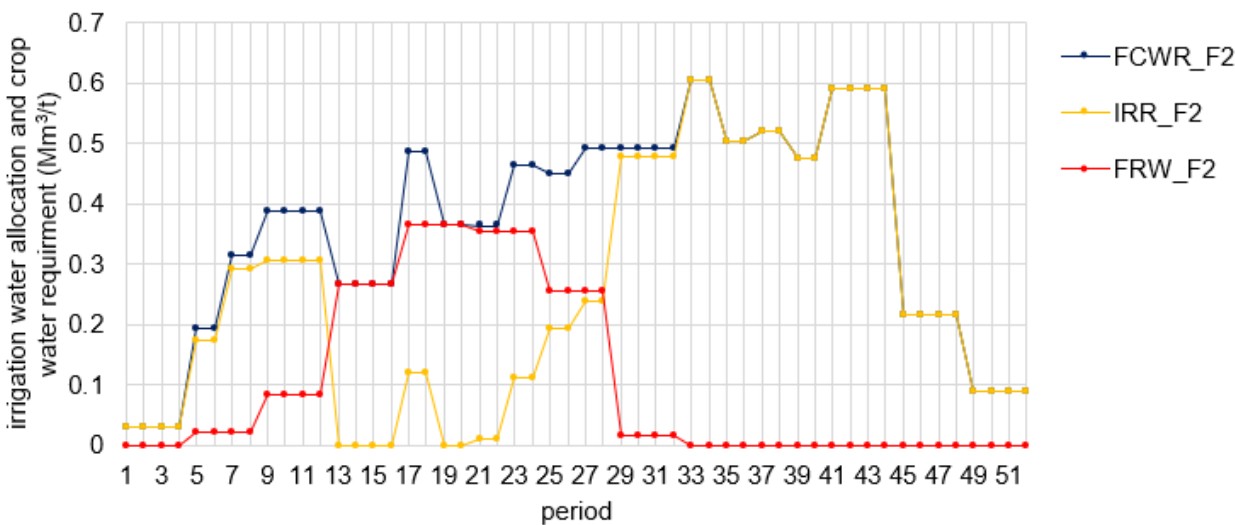

**Figure 17.** Allocated irrigation water (orange), crop water requirement (blue), and available rainfall (red) for farm 2 (Omo-Gibe river basin)(Mm³/week).

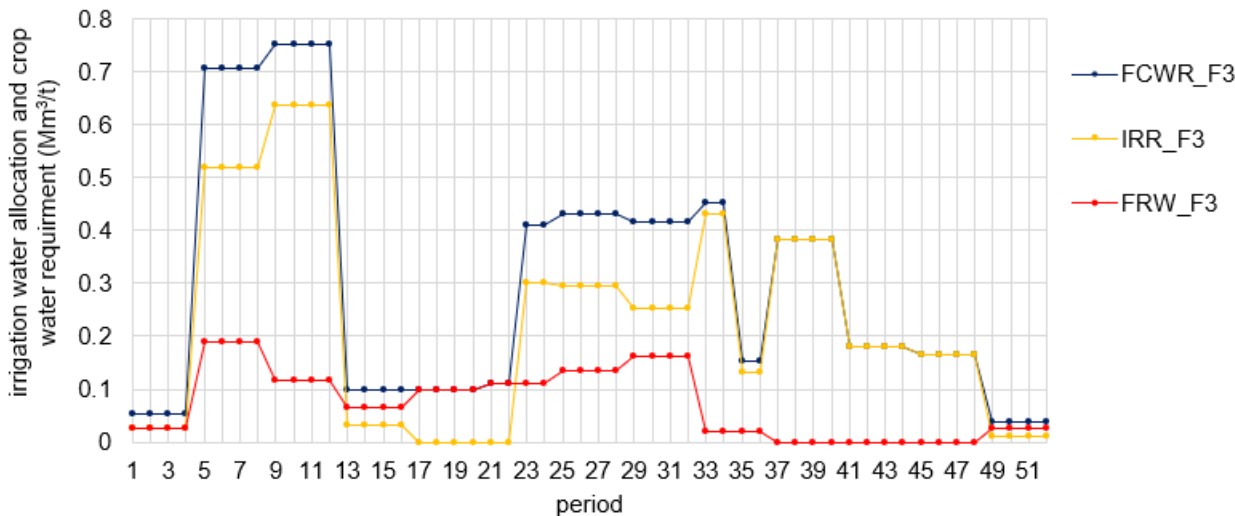

**Figure 18.** Allocated irrigation water (orange), crop water requirement (blue), and available rainfall (red) for farm 3 (Omo-Gibe river basin) (Mm³/week).

### 3.3. Land-Environment-Climate Nexus (LECN)

This section dealt with the interrelation among land use, environmental attribute, and climate change mitigation. It presented the optimal environmental impacts of land activities and the contribution towards soil organic carbon sequestration in reducing the greenhouse gas from the atmosphere. The allocated cropland has resulted in a certain amount of nitrate leaching and soil loss. These activities can negatively impact pollution and soil fertility. Any land use activities are expected to have some impact on natural resource management. However, there should be a certain threshold that must be considered. This study optimizes nitrate leaching, soil loss, and soil organic carbon sequestration from the allocated cropland. The excess nitrate leaching and soil loss are nil. Indirectly, it has met the maximum limit requirement of environmentally induced problems (Figure 19). Farm 3 has produced less nitrate leaching and soil loss compared with farm 1 and farm 2. This may be due to many factors, such as alternative crop succession allocation with less nitrate leaching and crop succession, etc. Farm 1 has used 90% of the land available, whereas farm 3 has used 100% of the available land. However, it has produced high nitrate leaching and soil loss. This shows that the less size of land allocation may or may not produce more environmentally induced problems. It relies on multiple factors, cropland management, crop landscape, types of crop rotation, fertilizer use, and many more. However, in the case of soil organic carbon sequestration, the more the land is allocated, the more the climate mitigation effect is due to more opportunity to soil organic carbon sequestration (Table 11). Farm 2 has sequestered more soil organic carbon than Farm 1, which has less allocated land size. In general, this section summarized the land allocation reaction to environmental and climate mitigation effects. It just only shows a one-way interaction. The model optimized the environmental and climate aspects of the allocated cropland. The reverse impacts of environmental and climate change mitigation aspects on land productivity are not included.

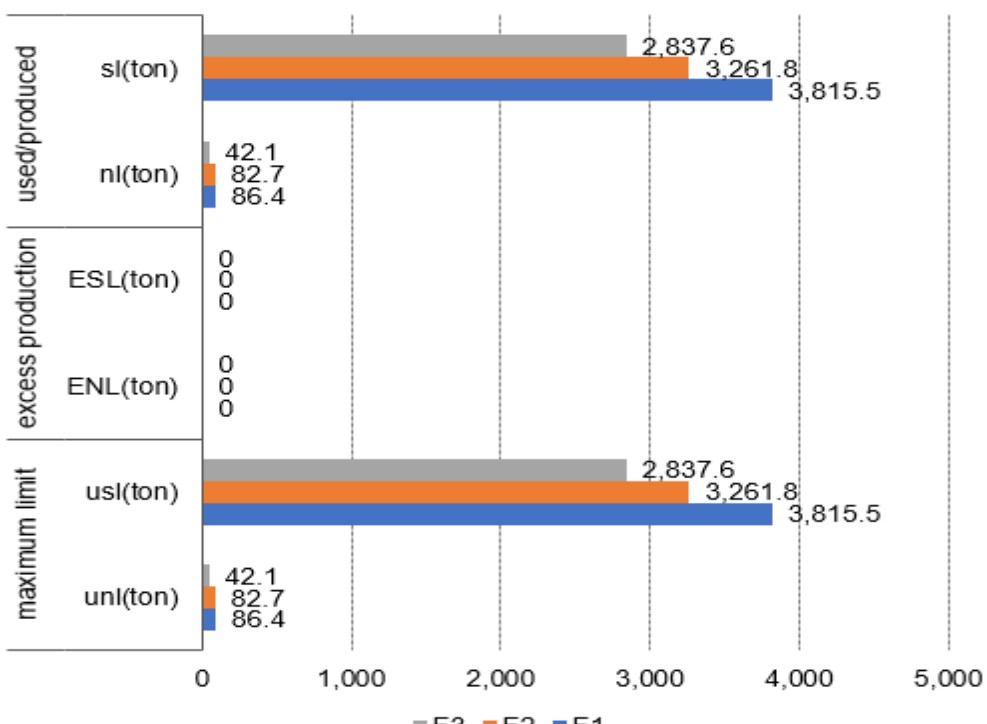

**Figure 19.** Environmental induced problems from allocated alternative crop successions per farm per year(ton/farm/year) (Farm 1, 2, and 3).

**Table 11.** Allocated soil organic carbon sequestration.

|  | **F1** | **F2** | **F3** |
|---|---|---|---|
| total soc sequestration (ton) | 465.3 | 491.7 | 449.5 |
| minimum limit (ton) | 465.3 | 491.7 | 449.5 |
| SSOC(ton) | 0 | 0 | 0 |

*3.4. Water-Energy-Water Supply-Ecosystem Nexus (WEEN)*

This referred to the linkage of water allocation for   energy, water supply, and ecosystem demands.

3.4.1. Reservoirs` volume and flow water model simulation results

Water from the three reservoirs,Gibe I, Gibe III, and Gibe IV, along the Omo-Gibe river basin is   allocated to four hydropower stations. The model simulation for the reservoir volume storage is presented in (Figures 20–22).

Gibe I Reservoir water volume storage and inflow/outflow model simulation results

Figure 20 shows the Gibe I reservoir volume storage characteristics. The reservoir volume falls below the minimum capacity requirement limit except in the middle and end of the modeling period. There is a penalty for unmet reservoir volume storage capacity. The reservoir inflow/outflow from and to the river link indicates that there is no reservoir outflow to the river link, but there is an intermittent reservoir inflow for a few weeks during the simulation period.

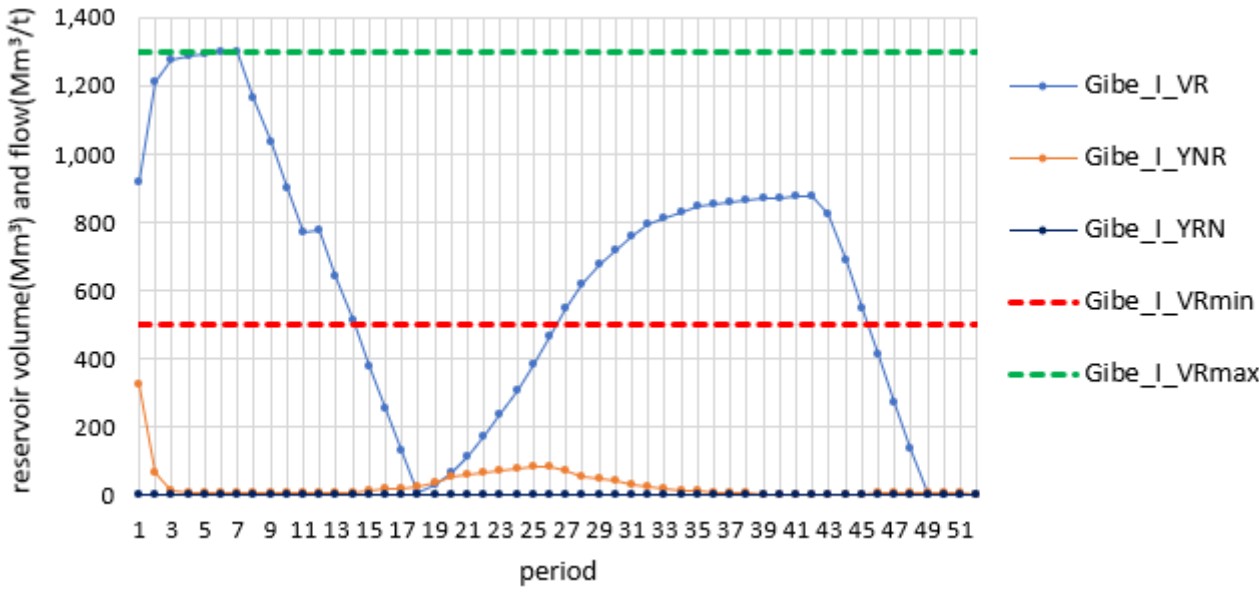

**Figure 20.** Gibe_I reservoir volume storage and reservoir inflow/outflow from and to river link (Mm³:Million cubic meters).

Gibe III Reservoir water volume storage and inflow/outflow model simulation results

Figure 21 presents the Gibe III reservoir model simulation. The pattern of the reservoir volume storage has been maintained throughout the simulation period. There is no reservoir outflow to the river link, but there is a slight inflow to the river link.

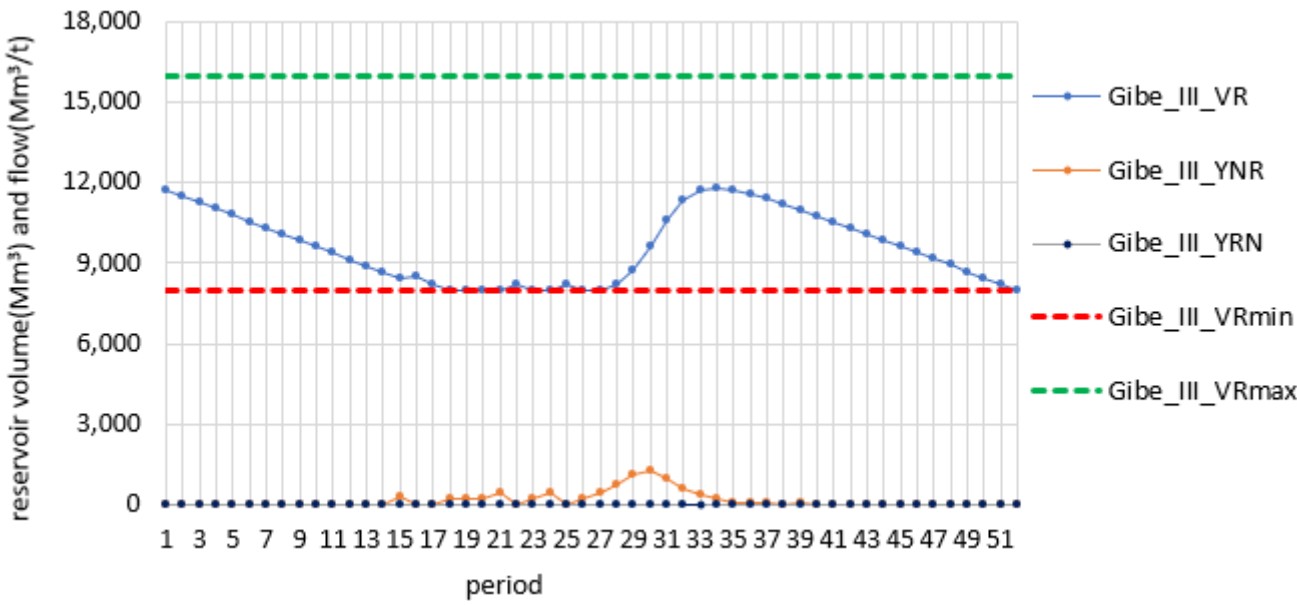

**Figure 21.** Gibe III reservoir volume storage and inflow/outflow from and to river link.

Gibe IV Reservoir storage volume and (i/o) from and to river link model simulation results

Figure 22 shows the simulation results for the Gibe IV reservoir volume storage and inflow/outflow. The pattern of the reservoir volume storage stays within the capacity limit throughout the simulation period. The reservoir outflow is zero, but there is a considerable amount of reservoir inflow.

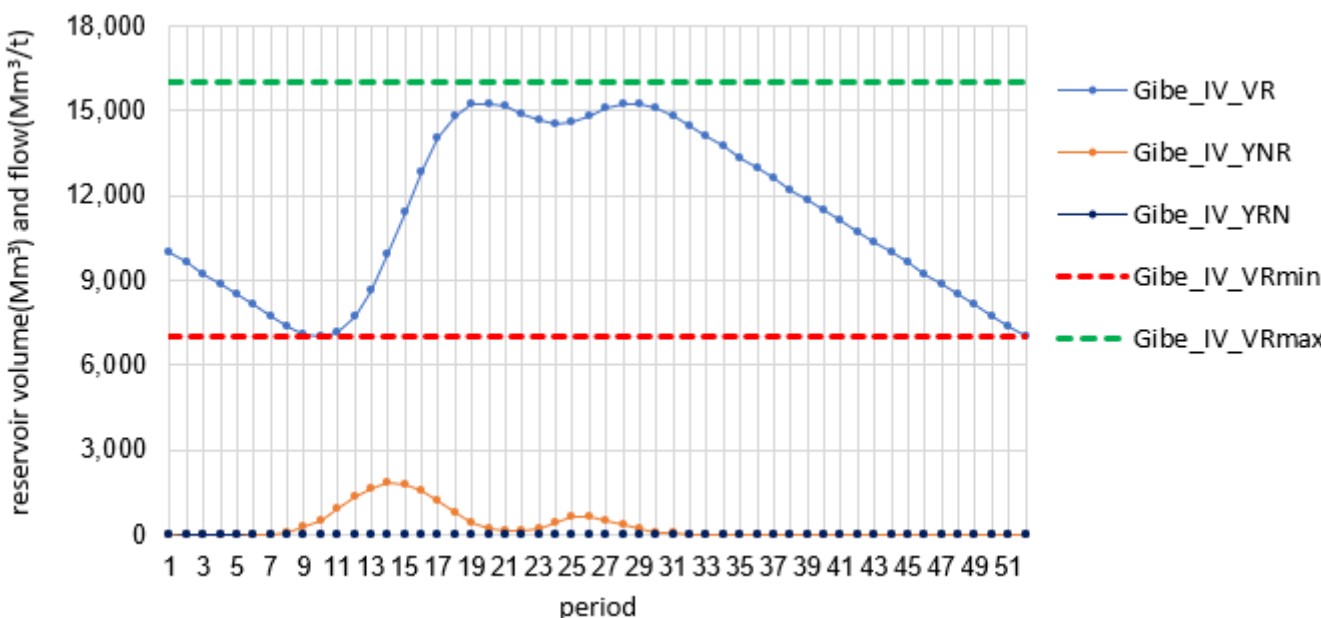

**Figure 22.** Gibe IV reservoir water volume storage and inflow/outflow model simulation results.

### 3.4.2. Water-Energy Nexus

Water is used to generate hydropower energy and to meet the water demands of agriculture and municipalities. In this regard, water is allocated to hydropower demand stations. This represents a one-way relationship between water-energy nexus. The model optimized and allocated water to hydropower energy demand sites (Gibe I,II,III and IV) along the Omo-Gibe cascade reservoirs.

The water allocation to the Gibe cascade hydropower stations has been met except for Gibe I and Gibe II (Figures 23 and 24). Gibe I and Gibe II have the same intake capacity. The water flows to the Gibe I hydropower station; after turning the turbine, the water continues through a long tunnel to the Gibe II hydropower station and then joins the Gibe river (Figures 2 and 3). In this study, there are unmet hydropower water demands for Gibe I and II hydropower stations. There is a penalty for unmet demands (Table 12). The water allocation demand for hydropower generation at Gibe III and IV has been fully met (Figures 25 and 26).

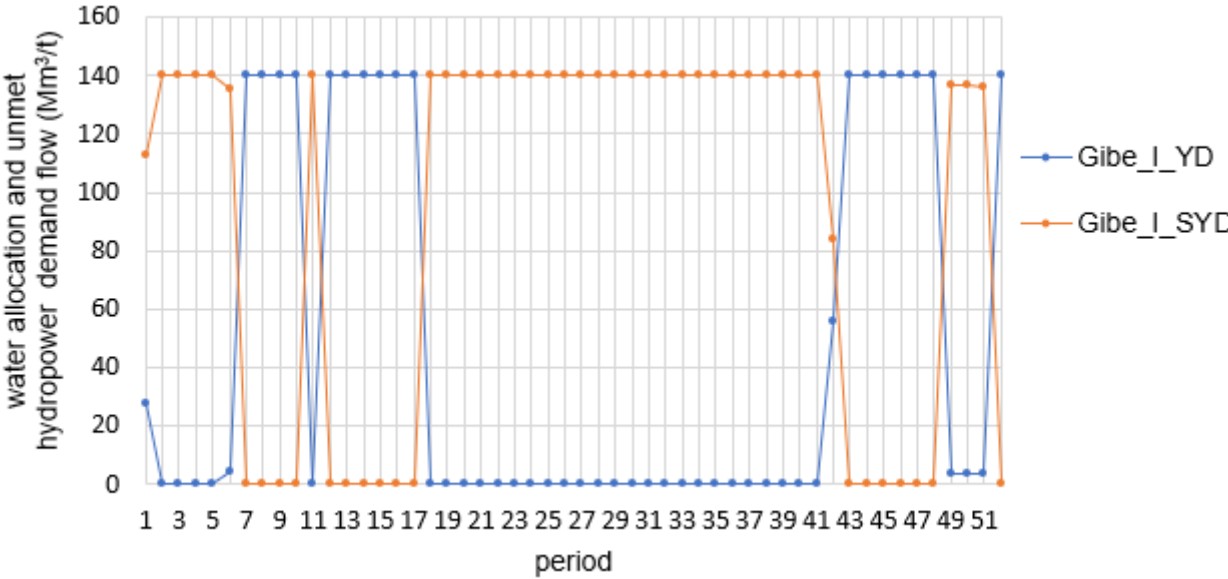

**Figure 23.** Water allocation and unmet demand water flow for the Gibe I hydropower station (Mm³/week).

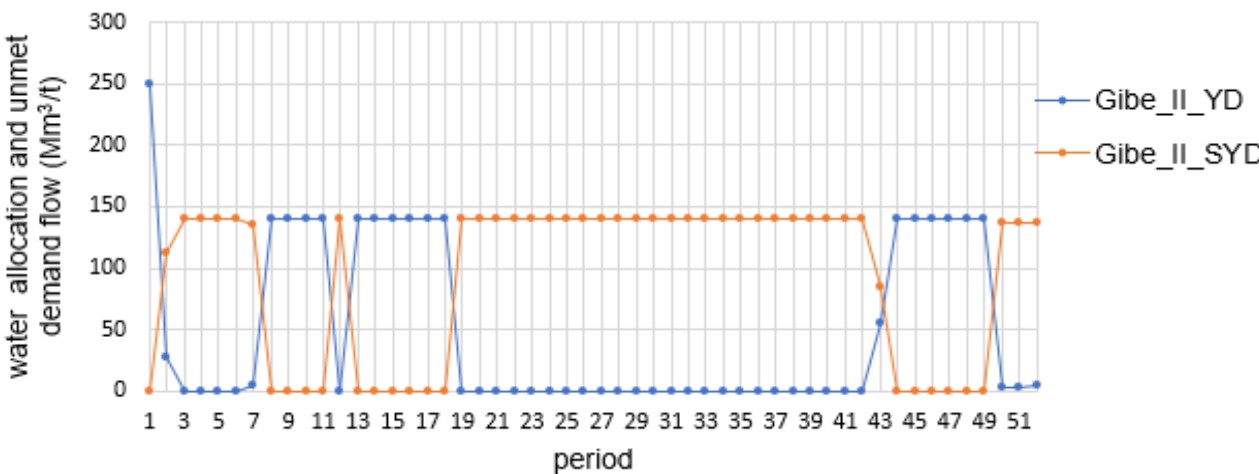

**Figure 24.** Water allocation and unmet demand water flow for the Gibe II hydropower station (Mm³/week).

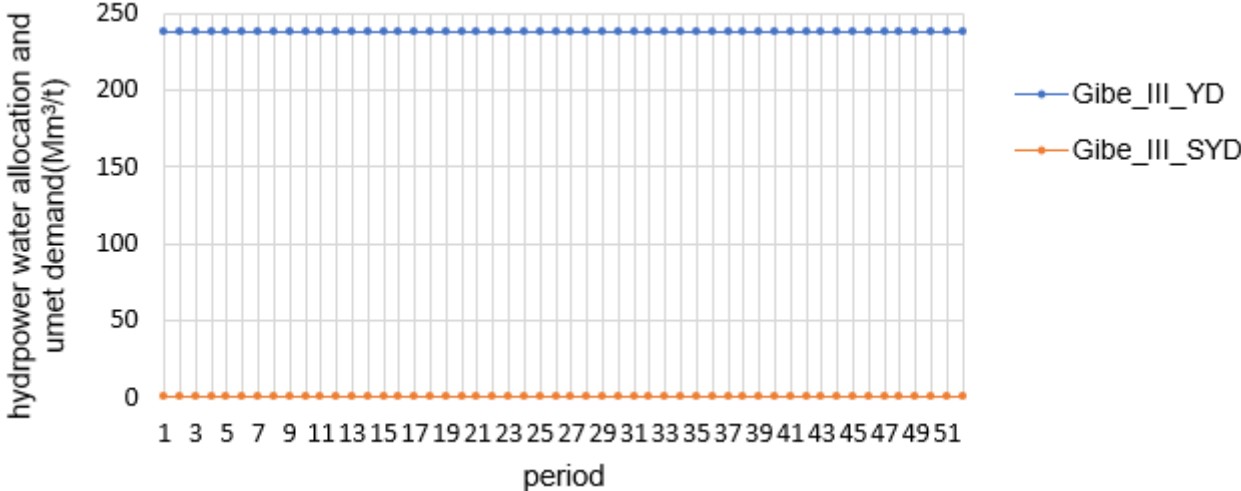

**Figure 25.** Water allocation and unmet demand water flow for the Gibe III hydropower station. (Mm³/week).

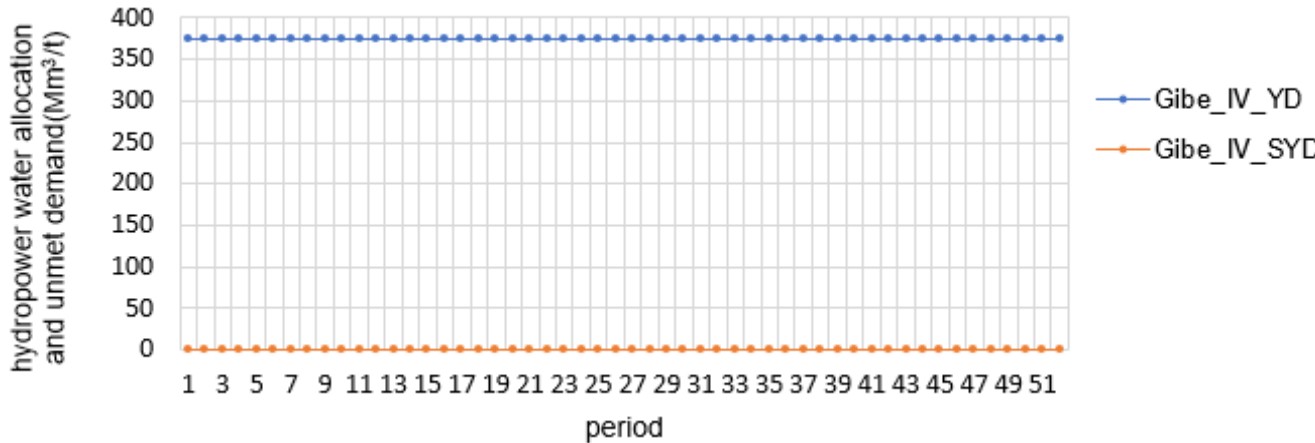

**Figure 26.** Water allocation and unmet demand water flow for the Gibe IV hydropower station (Mm³/week).

**Table 12.** Comparison of objective functions values under the initial, scenario I, and scenario II conditions.

| Condition | Optimal Value | Type | Income (Usd) ∑(ci*X) | Cost (Usd) ∑(cc*X) | Pr*(YNR + YRN) | | Pcal*SCAL | | Pssoc*SSOC | Penl*ENL | Pesl*ESL | Pvr*SVR | | Pd*SYD | |
|---|---|---|---|---|---|---|---|---|---|---|---|---|---|---|---|
| | | F1 | 1,659,808.4 | 462,988.8 | r1 | 42.6 | F1 | 0 | 0 | 0 | 0 | r1 | 25,000 | d1 | 960,314.5 |
| | | F2 | 1,787,219.9 | 483,776.9 | r2 | 250.6 | F2 | 0 | 0 | 0 | 0 | r2 | 0 | d2 | 960,314.5 |
| | | F3 | 1,785,441.7 | 388,814.7 | r3 | 482.6 | F3 | 0 | 0 | 0 | 0 | r2 | 0 | d3 | 0 |
| IC | | | | | | | | | | | | | | d4 | 0 |
| | | | | | | | | | | | | | | d5 | 0 |
| | | | 5,232,469.9 | 1,335,580.4 | | 775.9 | | 0 | 0 | 0 | 0 | | 25,000 | | 1,920,628.9 |
| | | | 5,232,469.9 | 3,281,985.1 | | 775.8 | | | | 1,945,628.9 | | | | | |
| | 1,950,485 | = | 1,950,485 | | | | | | | | | | | | |
| | | F1 | 1,729,933.6 | 462,988.8 | r1 | 42.3 | | 0 | 0 | 0 | 1844.3 | r1 | 25,000 | d1 | 963,258.6 |
| | | F2 | 1,777,964.3 | 464,965.7 | r2 | 250.6 | | 0 | 160.8 | 0 | 0 | r2 | 0 | d2 | 963,258.6 |
| | | F3 | 1,783,800 | 374,110 | r3 | 482.6 | | 0 | 52.9 | 6.9 | 0 | r3 | 0 | d3 | 0 |
| SCI | | | | | | | | | | | | | | d4 | 0 |
| | | | | | | | | | | | | | | d5 | 0 |
| | | | 5,291,697.9 | 1,302,064.4 | | 775.5 | | 0 | 213.7 | 6.9 | 1844.3 | | 25,000 | | 1,926,517.2 |
| | | | 5,291,697.9 | 1,302,064.4 | | 775.5 | | | | 1,953,582.2 | | | | | |
| | | | | | | | | | | 1,954,357.7 | | | | | |
| | 2,035,276. | = | 2,035,276 | -- | | | | | | | | | | | |
| Δ | ++ | | ++ | -- | | 0 | - | 0 | ++ | + | ++ | | 0 | | ++ |

|  |  |  |  |  |  |  |  |  |  |  |  |  |  |  |
|---|---|---|---|---|---|---|---|---|---|---|---|---|---|---|
| | ***84,794*** | +59,227.9 | −33,515.9 | | 0 | −0.3 | 0 | ++213.7 | +6.9 | ++1844.3 | | 0 | | ++5888.3 |
| | F1 | 1,178,491.7 | 300,942.7 | r1 | 43.1 | | 0 | 489.5 | | | r1 | 32,500 | d1 | 957,502.8 |
| | F2 | 1,274,938.2 | 314,455.0 | r2 | 371.9 | | 2744.7 | 593.9 | | | r2 | 95,704.2 | d2 | 957,502.8 |
| | F3 | 1,336,328.6 | 252,729.5 | r3 | 499.4 | | 0 | 477.4 | | | r3 | 76,503.6 | d3 | 0 |
| SCII | | | | | | | | | | | | | d4 | 0 |
| | | | | | | | | | | | | | d5 | 0 |
| | | 3,789,758.5 | 868,127.2 | | 914.4 | | 2744.7 | 1560.8 | 0 | 0 | | 204,707.7 | | 1,915,005.5 |
| | | 3,789,758.5 | 868,127.2 | | 914.4 | | | | 2,124,018.8 | | | | | |
| | 796698.1 | = | 796698.1 | | | | | | | | | | | |
| Δ | - | **-** | - | - | + | ++ | + | 0 | 0 | | ++ | 0 | -- |
| | | −1442711.5 | −467453.2 | | +138.6 | ++2744.7 | ++1560.8 | 0 | 0 | | ++179707.7 | | --5623.4 |

*IC(base/initial scenario), SCI(scenario I condition); SCII (scenario II condition), Δ change, F(Farm1,2,3). -- large decrease;- small decrease;++ large increase;+ small increase;0 no change; \* Gibe_near_Abelti town is the main river channel.*

### 3.4.3. Water-Water Supply Nexus

Water is allocated to towns near the upper Omo-Gibe river basin, located around Gibe I and II hydropower stations. The water supply demand for Deneba and Sokoru towns from river link six has been fully met in this optimization model (Figure 27).

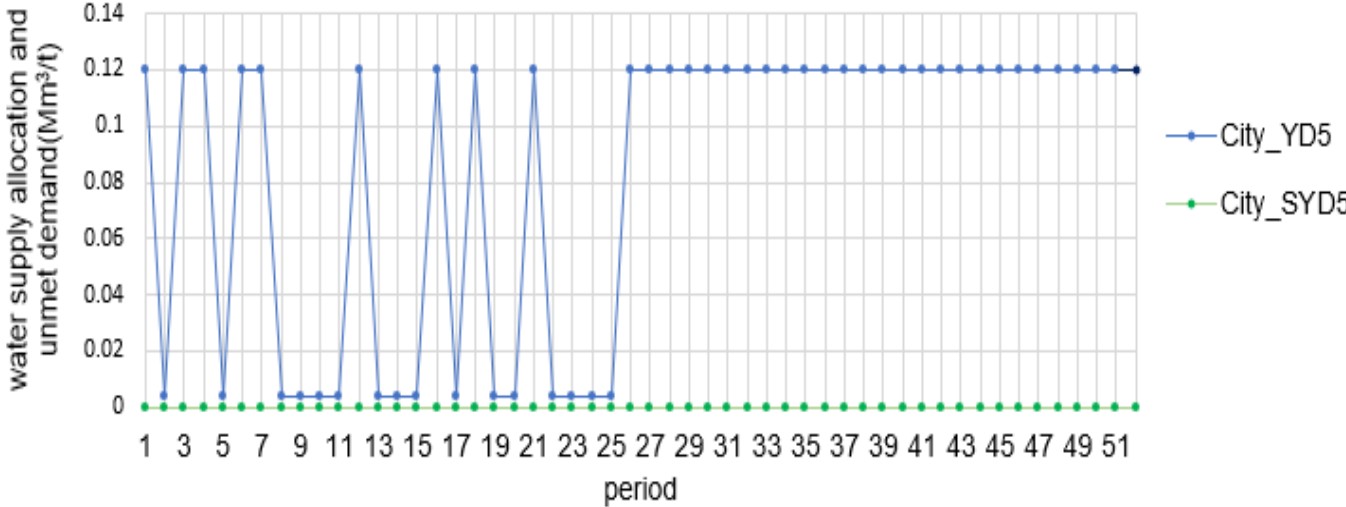

**Figure 27.** Water allocation and unmet demand flow for Sokoru & Deneba towns' municipal water supply (Mm$^3$/week).

### 3.4.4. Water-Ecosystem Nexus

When river water flows in a river channel, there should be a minimum water flow and volume. In this study, the minimum river flow and volume demands have been guaranteed (Figures 28 and 29). This supports the preservation of river biodiversity and conservation of habitats dependent on the Omo-Gibe river flow regime. It reduces the impact of river flow shortage on the river biota (e.g., fish population, fauna, and so on). The river flow pattern shows there is a high fluctuation. The high peak is observed in link 7, which has a relatively high inflow from the Gilgel Gibe river and Gibe II hydropower station. It is one of the biggest tributaries of the Omo-Gibe river, which is located after Gibe I and Gibe II hydropower stations, so it has a high peak flow, and the peak is slightly decreased as the water flows to the next river link. The other tributaries' inflow effect on peak flow is minimum because they are nearby either the irrigation or hydropower demand sites. Their peak flow effect is reduced due to river flow diversion or abstraction to different demand sites. However, the volume pattern has a slightly different pattern due to the four tributaries that flow into the Omo Gibe river. For example, the volume in river link 13 shows a high incline volume graph. This is because of the major tributary so-called Gojeb river. It is one of the biggest tributaries of the Omo-Gibe river between Gibe II and Gibe III. River link six, indicates a constant volume throughout the model simulation period. It is the river link after Gibe I. This is because the difference between the inflow-minus-outflow is nearly zero or minimum. The outflow from the river link to the Gibe I reservoir is compensated by the inflow to the river link six by the Gilgel Gibe river and by the return flow from the Gibe II hydropower station. So, there is little significant difference between inflow and outflow. Further, this study has allocated a certain amount of river flow to the downstream areas with a minimum of 8.5 Mm$^3$ and a maximum limit of 34 Mm$^3$ per week. This prevents unexpected conflicts due to water shortages and helps preserve the downstream river ecosystem (Figure 30).

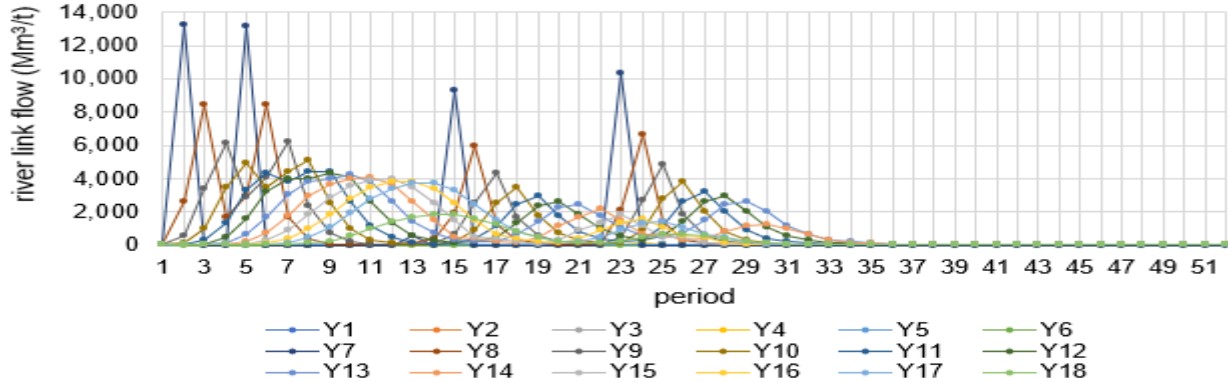

**Figure 28.** Omo-Gibe river link flow model simulation results (Mm³/week).

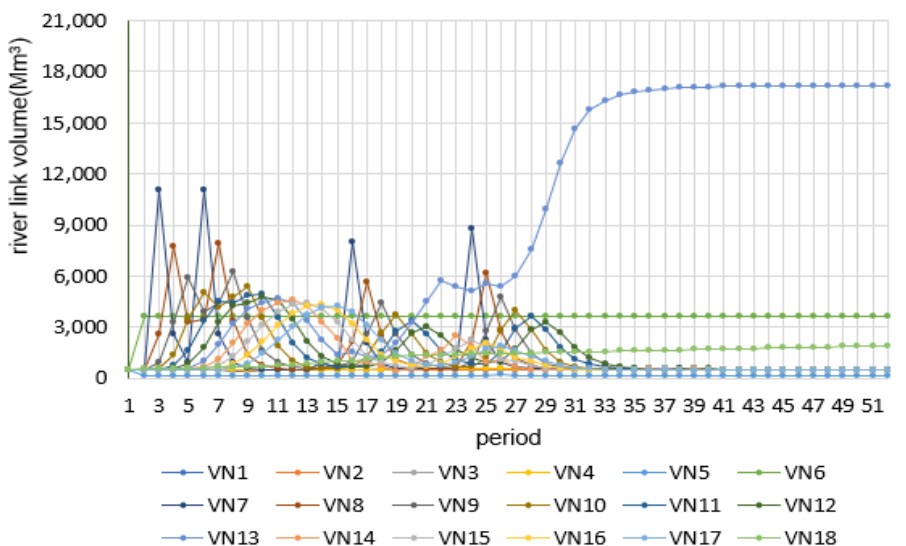

**Figure 29.** Omo-Gibe river volume model simulation results.

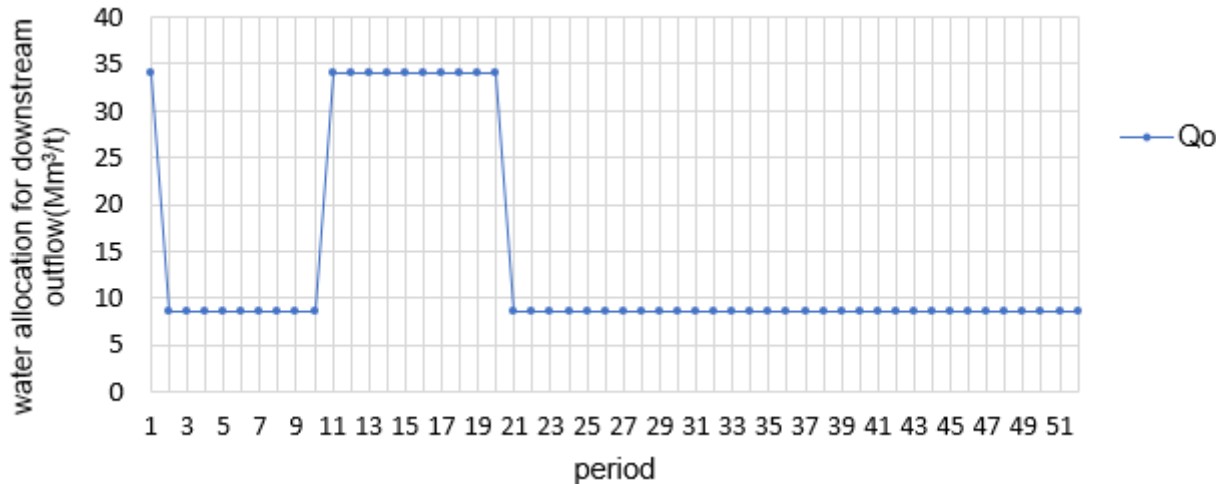

**Figure 30.** Water allocation for the downstream river flow model simulation (Mm³/week).

*3.5. Scenario Based Analysis for the Omo-Gibe Iver Basin*

3.5.1. Scenario I

This scenario assumes no rainfall during the entire modeling period due to drought conditions (rw = 0, in farm 1, 2, and 3).

The land allocation pattern and size showed a slight change under the no rainfall condition compared to the initial model condition (Figure 31). An additional 101.3 ha of land is added. Despite no rainfall, sufficient flow of water supplements irrigation water demand. However, the increased land allocation resulted in more nitrate leaching and soil loss. It produced excess soil loss in farm 1, around 230.5 tons, and excess nitrate leaching in farm 3, which is more than 1.2 tons. Therefore, there is a penalty for it (Table 12). Figure 32 illustrates the irrigation water allocation comparison to the initial model condition. As there is no supplemental rain for crop productivity, more irrigation water is allocated to meet the crop water requirement per farm. There is a discrepancy in the allocated water allocation per farm per unit of time due to a slight change in the allocated pattern of alternative crop successions.

|    |    | P1 | P2 | P3 | P4 | P5 | P6 | P7 | P8 | P9 | P10 | P11 | P12 | P13 | P14 | P15 | P16 | P17 | P18 | P19 | P20 |
|----|----|----|----|----|----|----|----|----|----|----|-----|-----|-----|-----|-----|-----|-----|-----|-----|-----|-----|
|    | L1 | 0 | 0 | 0 | 0 | 205 | 0 | 0 | 0 | 95 | 0 | 0 | 0 | 0 | 0 | 0 | 0 | 0 | 0 | 0 | 0 |
|    | L2 | 0 | 0 | 0 | 0 | 0 | 0 | 0 | 300 | 0 | 0 | 0 | 0 | 0 | 0 | 0 | 0 | 0 | 0 | 0 | 0 |
| F1 | L3 | 0 | 0 | 0 | 0 | 0 | 0 | 0 | 144 | 127 | 29 | 0 | 0 | 0 | 0 | 0 | 0 | 0 | 0 | 0 | 0 |
|    | L4 | 0 | 0 | 0 | 0 | 0 | 0 | 0 | 0 | 300 | 0 | 0 | 0 | 0 | 0 | 0 | 0 | 0 | 0 | 0 | 0 |
|    | L1 | 0 | 0 | 0 | 0 | 0 | 0 | 0 | 259 | 0 | 41 | 0 | 0 | 0 | 0 | 0 | 0 | 0 | 0 | 0 | 0 |
|    | L2 | 0 | 0 | 0 | 0 | 0 | 0 | 0 | 0 | 0 | 300 | 0 | 0 | 0 | 0 | 0 | 0 | 0 | 0 | 0 | 0 |
| F2 | L3 | 0 | 0 | 0 | 0 | 0 | 0 | 221 | 0 | 0 | 0 | 0 | 0 | 0 | 79 | 0 | 0 | 0 | 0 | 0 | 0 |
|    | L4 | 0 | 0 | 0 | 0 | 0 | 0 | 0 | 0 | 0 | 0 | 0 | 0 | 0 | 0 | 0 | 0 | 300 | 0 | 0 | 0 |
|    | L1 | 0 | 0 | 0 | 0 | 0 | 0 | 0 | 0 | 0 | 300 | 0 | 0 | 0 | 0 | 0 | 0 | 0 | 0 | 0 | 0 |
|    | L2 | 0 | 0 | 0 | 0 | 0 | 0 | 0 | 0 | 0 | 300 | 0 | 0 | 0 | 0 | 0 | 0 | 0 | 0 | 0 | 0 |
| F3 | L3 | 0 | 0 | 0 | 0 | 0 | 0 | 300 | 0 | 0 | 0 | 0 | 0 | 0 | 0 | 0 | 0 | 0 | 0 | 0 | 0 |
|    | L4 | 0 | 0 | 0 | 0 | 0 | 0 | 300 | 0 | 0 | 0 | 0 | 0 | 0 | 0 | 0 | 0 | 0 | 0 | 0 | 0 |

**Figure 31.** Allocated alternative crop succession to the land unit under scenario I condition (ha).

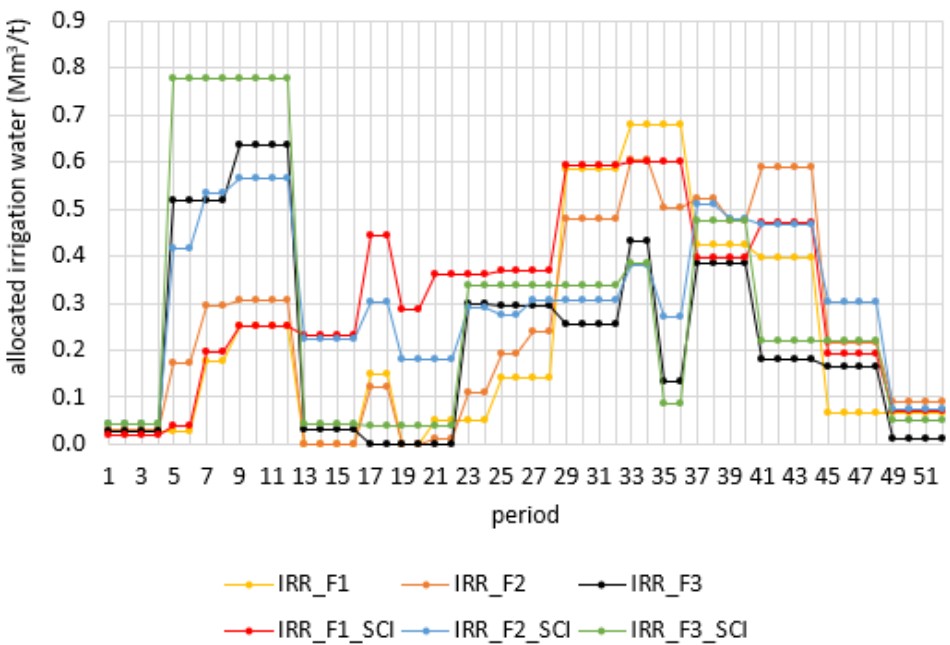

**Figure 32.** Comparison of irrigation water allocation between scenario I and base scenario model conditions(Mm³/week).

### 3.5.2. Scenario II

In this scenario, the crop budget (cost for labor, fertilizer, pesticides, etc.) is 35% lower compared to the base scenario condition (Farm 1, 2, 3), and the minimum volume storage capacity of the reservoirs is 30% higher (Gibe I, III and IV).

Figure 33 presents the allocation of alternative crop successions to land units under scenario II. The total size of land has decreased by 660.2 ha, which is around 20%. This has contributed to an unmet food calorie of 305 MegKcal in Farm 2 and also resulted in a high amount of unmet soil carbon sequestration of 312.2 tons (Table 13). The unmet food calorie in farm 2 is associated with a high minimum demand limit. It has a high minimum demand than farm 1 and 2. The irrigation water allocation pattern is different from the initial base condition. Furthermore, the allocation of irrigation water has decreased due to the allocation of smaller land sizes and changes in alternative crop successions allocations (Figure 34).

|   |   | P1 | P2 | P3 | P4 | P5 | P6 | P7 | P8 | P9 | P10 | P11 | P12 | P13 | P14 | P15 | P16 | P17 | P18 | P19 | P20 |
|---|---|---|---|---|---|---|---|---|---|---|---|---|---|---|---|---|---|---|---|---|---|
|    | L1 | 34 | 0 | 0 | 0 | 0 | 0 | 0 | 266 | 0 | 0 | 0 | 0 | 0 | 0 | 0 | 0 | 0 | 0 | 0 | 0 |
|    | L2 | 0 | 0 | 0 | 0 | 0 | 0 | 0 | 0 | 0 | 0 | 0 | 0 | 0 | 0 | 0 | 0 | 0 | 0 | 300 | 0 |
| F1 | L3 | 0 | 0 | 0 | 0 | 0 | 0 | 0 | 0 | 0 | 0 | 0 | 0 | 0 | 0 | 0 | 0 | 0 | 0 | 6 | 0 |
|    | L4 | 0 | 0 | 0 | 0 | 0 | 0 | 0 | 167 | 0 | 0 | 0 | 0 | 0 | 0 | 0 | 0 | 0 | 0 | 133 | 0 |
|    | L1 | 300 | 0 | 0 | 0 | 0 | 0 | 0 | 0 | 0 | 0 | 0 | 0 | 0 | 0 | 0 | 0 | 0 | 0 | 0 | 0 |
|    | L2 | 249 | 0 | 0 | 0 | 0 | 0 | 0 | 0 | 0 | 0 | 0 | 0 | 0 | 0 | 0 | 0 | 0 | 0 | 0 | 0 |
| F2 | L3 | 0 | 0 | 0 | 0 | 0 | 0 | 0 | 0 | 0 | 0 | 0 | 0 | 0 | 0 | 0 | 0 | 0 | 0 | 76 | 0 |
|    | L4 | 0 | 0 | 0 | 0 | 0 | 0 | 0 | 0 | 0 | 0 | 0 | 0 | 0 | 0 | 0 | 0 | 300 | 0 | 0 | 0 |
|    | L1 | 0 | 0 | 0 | 0 | 0 | 0 | 0 | 0 | 0 | 108 | 0 | 0 | 0 | 0 | 0 | 0 | 0 | 0 | 0 | 0 |
|    | L2 | 5 | 0 | 0 | 0 | 0 | 0 | 0 | 0 | 0 | 295 | 0 | 0 | 0 | 0 | 0 | 0 | 0 | 0 | 0 | 0 |
| F3 | L3 | 0 | 0 | 0 | 0 | 0 | 0 | 0 | 0 | 0 | 0 | 0 | 0 | 0 | 0 | 300 | 0 | 0 | 0 | 0 | 0 |
|    | L4 | 0 | 0 | 0 | 0 | 300 | 0 | 0 | 0 | 0 | 0 | 0 | 0 | 0 | 0 | 0 | 0 | 0 | 0 | 0 | 0 |

**Figure 33.** Allocated alternative crop successions to land unit under scenario II conditions (ha).

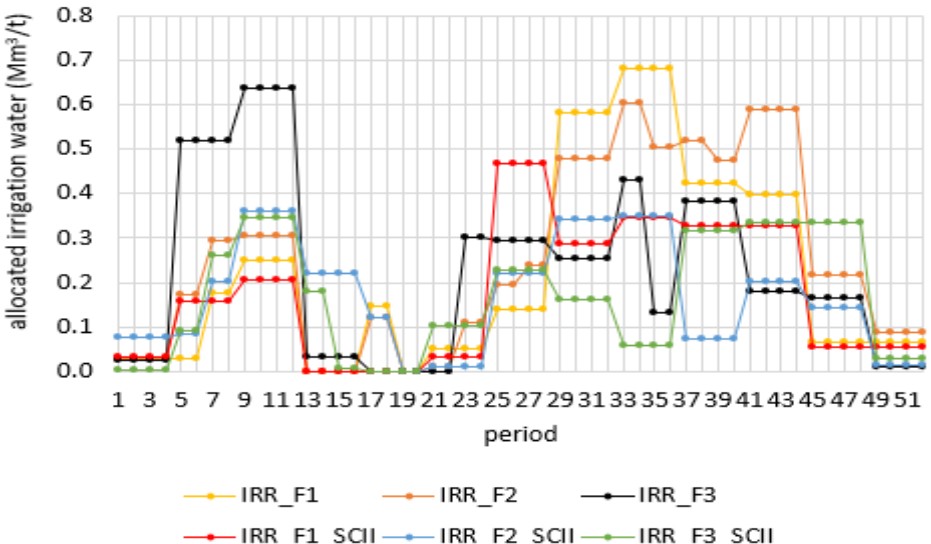

**Figure 34.** Comparison of irrigation water allocation between scenario II and the initial base scenario conditions (Mm³/week).

**Table 13.** Comparison of optimized land allocation and attributes under different model scenario conditions.

| Type | Farm | Area (ha/Year) | Food Demand | | Environmental Maximum Limit (ton/Year) | | Climate Change Mitigation Demand (ton/Year) |
|---|---|---|---|---|---|---|---|
| | | | Food Calorie Production (MegKcal/Year) | | Nitrate Leaching | SOIL LOSS | Unmet |
| | | | Excess | Unmet | Excess | Excess | |
| | | | $ECAL_f$ | $SCAL_f$ | $ENL_f$ | $ESL_f$ | $SSOC_f$ |
| Base scenario(IC) | F1 | 1098.7 | 295.4 | 0 | 0 | 0 | 0 |
| | F2 | 1200 | 615.6 | 0 | 0 | 0 | 0 |
| | F3 | 1200 | 563.3 | 0 | 0 | 0 | 0 |
| total | | 3498.7 | 1474.2 | 0 | 0 | 0 | 0 |
| SC_I | F1 | 1200 | 0 | 0 | 0 | 230.5 | 0 |
| | F2 | 1200 | 0 | 0 | 0 | 0 | 32.2 |
| | F3 | 1200 | 272 | 0 | 1.2 | 0 | 10.6 |
| total | | 3600 | 272 | | 1.2 | 230.5 | 42.8 |
| change | | + | + | 0 | 0 | 0 | 42.8 |
| Δ | | ++101.3 | | 0 | +1.2 | +230.5 | +42.7 |
| SC_II | F1 | 905.6 | 51.1 | 0 | 0 | 0 | 97.9 |
| | F2 | 924.4 | 0 | 305 | 0 | 0 | 118.8 |
| | F3 | 1008.5 | 2051.4 | 0 | 0 | 0 | 95.5 |
| total | | 2838.5 | 2102.5 | 0 | 0 | 0 | 312.2 |
| change | | - | + | + | 0 | 0 | + |
| Δ | | --660.2 | ++628.3 | +305 | 0 | 0 | +312.2 |

*IC(base/initial scenario), SCI(scenario I condition); SCII (scenario II condition), Δ change, F(Farm1,2,3); -- large decrease;- small decrease;++ large increase;+ small increase;0 no change.*

### 3.5.3. Scenario III

When energy security demand and reservoir technical capacity costs increase.

This scenario considered that the demand for energy security and reservoir minimum volume storage limit increase, i.e., when the penalty for unmet energy demand and reservoir volume minimum limit increase by 50% compared to the base scenario.

This is intended to understand how the overall optimum economic benefits respond to changing nexus demand security conditions. For example, when water for energy demand security increased by 50%, the overall optimum economic benefits from the land and water allocation decreased by more than 50% (i.e., from 1950485 to 990170.4 USD). When the technical demand security for maintaining reservoir minimum volume limit increased by 50%, the optimum net benefit decreased by 0.6% (i.e., from 1950485 to 1937985 USD). This shows that energy demand security has incurred a high cost on the overall economic net return from combined land and water resources allocations. This result shows that under a competing land and water resources use condition. It is very challenging and complex to maintain and sustain energy demand and other essential ecosystem services without optimal natural resource allocation.

### 3.6. Comparisons of Objective Functions Sensitivity Analysis

Objective Functions Analysis

Table 12 describes the objective function analysis comparison of the model's initial base scenario output with projected scenario conditions. The optimal objective value increased under the scenario I condition by 84,791 USD. This means more crop income (59,227.9 USD) is generated. This is attributed to a relatively large land size allocation in scenario I compared with the initial condition. The shortage of rainwater is compensated by the sufficient availability of river water. So more water from the river is allocated to meet the crop water requirements. However, the allocation of more river water to meet

the irrigation demand has increased penalty costs from the unmet hydropower water demand of Gibe I and Gibe II hydropower stations. It has an extra penalty cost of 5888.3 USD. This is due to less reservoir water allocation to Gibe I and Gibe II. Gibe II takes the same amount of water flow from Gibe I intake since it is connected by the underground tunnel. Eventually, the water returns to the Omo-Gibe river channel and goes downstream.

The objective function compared with the scenario II condition has different results than scenario I. In this case, the optimal objective function value decreased by 796,698.1 USD. This is connected with less crop budget availability, and it has resulted in less crop income (1,442,711.5 USD). High penalty cost is incurred due to unmet reservoir volume storage due to reservoir volume storage increase by 30% (Table 12). On the contrary, the penalty cost for unmet hydropower demands of Gibe I and Gibe II has decreased by 5623.4 USD.

## 4. Discussions

The LWFEEEECN approach is a complex and sophisticated one. It requires large data sets and parameters, including agricultural production, hydrological, meteorological, socio-economic, and environmental demand limits. For example, in this study, the land part alone needs about 99,840 data points for spatial and temporal land inputs. Extensive data is also required for the water part. It is very challenging to obtain the exact real data for each input set. We used close approximate data inputs to run the model and obtain a realistic depiction of the Omo-Gibe river basin. Allocations and demands are analyzed with consideration of cascade hydropower sites such as Gibe I, Gibe II, Gibe III, and Gibe IV. Three communal potential farm sites are considered: two are located on the upstream sides, and one is on a gently sloping agricultural area at the lower end. Water supply for Deneba and Sokoru towns is treated as one-demand node because they are near each other and likely draw water from the same river segment. Land and water resources are essential resources for human and natural ecosystem services. The competitions for these scarce resources are at high stake. There is increasing pressure to meet the demand for land, water, food, energy, environment conservation, river ecosystem, and climate change mitigation [48]. Therefore, there is a strong interdependence between LWFEEN interactions. Such nexus requires unbiased decisions without treating each factor independently. Indirectly, this issue should be addressed concertedly. Implementing an optimization model using the LWFEEN approach would provide a sustainable decision and improve the efficiency of natural resources management of the Omo-Gibe river basin. An optimization method is a dominant tool for addressing complex nexus issues [49].

This study's Gebre optimization model met the demands and requirements for integrated water and land resource allocation. However, there are some unmet demands which are compensated for with penalties. For example, the hydropower water demand for Gibe I and Gibe II is not fully met, but there is incurred penalty cost on the objective function, which decreases the maximum net profit. However, the hydropower water demand for Gibe III and Gibe IV is met throughout the model simulation period. The water supply demand for Deneba and Sokoru towns is also met. The irrigation water allocation is dependent on the allocation of alternative crop successions to land units and rainfall availability. The alternative crop succession allocation is closely correlated to multiple constraints of land productivity and environmental conservation limits. Despite all, the irrigation water demand of each farm has been met. More food calories are produced from allocated croplands under climate-smart land use planning. In conclusion, the applied Gebre optimization model has solved conflicting constraints, trade-offs, and synergies existing among land-water-food-energy-environmental conservation under the context of climate change mitigation and river ecosystem-ecosystem services (LWFEEN) in the Omo-Gibe river basin.

Previously studies have been done by Sundin [17] on exploring the water-energy nexus in the Omo river basin. However, the study used a combination of the hydrological

model (Topkapi-ETH) and OSeMoSYS to generate energy from reservoir water availability received from the hydrological model. It applied the model for each Gibe cascade reservoir separately. Besides, the study report shows that the coupling was incomplete in generating hydropower energy based on the reservoir's hydrological characteristics. This exploration did not include the whole cascade reservoirs concurrently and was unable to allocate water to each demand node. More studies have been conducted to address the interaction between different demand sectors. For example, Li et al. [50] used an optimization model for sustainable bioenergy production considering the energy-food-water-land nexus in a region of northeast China. However, the research has concentrated on trade-offs of the interaction between economic and environmental impacts on bioenergy production. It did not explicitly use land and water resources in its model application. Rather, it simply considered land policy and water supply demand as constraints. Many more optimization model studies have been conducted extensively on the nexus approach [12,50,51]. However, none of them have implemented a detailed integrated land and water resources network in their model applications. Furthermore, they did not include multiple factors in their nexus approach. This study has embedded multiple factors in the complex nexus approach and solved multiple demands with optimal solutions.

## 5. Conclusions

Land and water resources management in a basin facing severe water-energy-food nexus demands information-based management due to the growing demand for food and ecosystem services. The Gebre-tool allows decision-makers to address natural resource allocation problems through strategic policies and to meet sustainable development goals. This model is applied in the Omo-Gibe river basin, Ethiopia, to address complex multiple-sector interactions.

The results illustrated that it is possible to optimize land and water resources while meeting demands from various stakeholders. The increase or decrease of a certain factor cannot be achieved without affecting the objective functions. The model has demonstrated the competition between land and water. For instance, as cropland increases, competition for irrigation water arises, and similarly, it influences water demand for hydropower, water supply, and river ecosystem. There is high competition and interaction among onsite land eco-services. An increase in land productivity leads to rising environmental problems, affecting sustainable agricultural land management. Overall, this study succeeded in proposing the optimal allocation of the limited natural resources through the land-water-food-energy-environment nexus (LWFEEN) under climate change mitigation and river ecosystem service approach in the Omo-Gibe river basin. It enhanced the understanding of the multiple sector interactions. This optimization tool is flexible and versatile and can be applied geographically worldwide to address natural resource allocation problems to meet the growing demands for ecosystem services, including nature conservation. Moreover, this tool accommodates temporal-spatial-based large data sets which can be applied at any scale.

This research study can be extended by including additional nexus sectors, increasing temporal resolution, and considering two-way nexus interactions or the impacts of environmental problems on land and water allocation and vice versa.

**Author Contributions**: Conceptualization, S.L.G., J.V.O. and D.C.; Data curation, S.L.G., J.V.O. and D.C.; Formal analysis, S.L.G., J.V.O. and D.C.; Funding acquisition, J.V.O. and D.C.; Investigation, S.L.G.; Methodology, S.L.G., J.V.O. and D.C.; Project administration, D.C.; Resources, J.V.O. and D.C.; Software, S.L.G.; Supervision, J.V.O. and D.C.; Validation, S.L.G., J.V.O. and D.C.; Visualization, S.L.G.; Writing—original draft, S.L.G.; Writing—review and editing, S.L.G., J.V.O. and D.C. All authors have read and agreed to the published version of the manuscript.

**Funding:** This research received no external funding.

**Data Availability Statement:** The model algorithm and the data used in this study are available from the corresponding author upon reasonable request.

**Acknowledgments**: This research was financially supported by the Global Mind`s Ph.D. scholarship provided by the KU Leuven University, Belgium.

**Conflicts of Interest**: The authors declare that they have no known competing financial interests or personal relationships that could have appeared to influence the work reported in this paper.

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
