# Peer review of "Optimizing the Combined Allocation of Land and Water to Agriculture in the Omo-Gibe River Basin Considering the Water-Energy-Food-Nexus and Environmental Constraints"

_land, doi:10.3390/land12020412_

Round 1

Author Response

Response to Reviewer1`s comments and suggestions

Review of Manuscript ID= land-2188810

The paper is concerned about how to maximize the economic benefit of land and water resources via resource allocation optimization model framework of land-water-food-energy-environment nexus under climate/ecosystem services (LWFEEN). In other word, the manuscript presents how wisely use our scarce resources that are land and water in Omo-Gibe River basin, Ethiopia with the objective function to maximize the economic benefit of land and water through the help of optimization model that is LWFEEN, which is interesting. It is quite relevant work and proud to inform you that I believe in the research you have undertaken. However, I do have some minor concerns that has to be clarify.

Please respond attentively to my comments and questions.

  • Please also correct the space problem in the introduction part like, the space before “Several mathematical decision-making tools have been developed and used in a different land and water resource allocation problems to address water-food-energy demands” and others.

Response

Dear reviewer, thank you for the comment.

The issue is corrected. All the space problems are considered throughout the manuscript.

  • In the introduction part line 6 and 7, “Land and water are scarce natural resources, of which, due to population in-crease, urbanization, and industrialization, the demand has increased.” Better to refresh it as “Land and water are scarce natural resources and their demand has increased due to population increase, urbanization, and industrialization.” Or …

Response

 Dear reviewer thank you for the comment.

 The sentence is corrected accordingly and included in the manuscript.

“Land and water are scarce natural resources, and their demand has increased due to  population growth, urbanization, and industrialization."

  • How is LWFEEN (water-energy-food-environment nexus) model different from other optimization models, how do the authors prefer to use this model rather than others?

Response

Dear reviewer thank you for the comment on the confusion between the Gebre model and the LWFEEN interactions approach. Please below are some of the explanations and this issue is considered in the manuscript.

  • The LWFEEN is not a model; it is an acronym to express i.e. the interaction between land, water, food, energy, and environment. The Gebre model optimizes the interactions of these components.
  • The Gebre-model term is introduced in the abstract…the concept is to apply the Gebre optimization model to address the interconnected issues of land-water-food-energy-environment (LWFEEN) under the context of climate change mitigation and river ecosystem services.
  • The abstract part is revised to avoid confusion between the Gebre model and the LWFEEN issues.
  • The Gebre model is an innovative and unique model that combines land and water allocation. It introduces the concept of allocating alternative crop successions to land units, which are dynamically linked to a river with a reservoir system. This type of integrated optimization model is rarely used or studied in the literature; thus the goal is to develop such kind of model and apply it to address the complex interactions between land, water, food, energy, and the environment in the context of climate change mitigation and river ecosystem management.
  • In the abstract the authors have said to implement the LWFEEN (land-water-food-energy-environment nexus) model to maximize land and water resources, in the other hand in the introduction part the authors have said the main objective was to apply the Gebre-model to the Omo-Gibe River basin in Ethiopia. So, what is the different between LWFEEN model and Gebre-model? Why you want to mention bothIf the objective is to apply Gebre-model why you want to mention LWFEEN? Is Gebre-model a part of LWFEEN?

Response

Dear reviewer thank you again for the comment.

This part is addressed and the abstract is completely revised. Please see the abstract section.

  • “In general, this model contributes to the concept of integrated land and water resource planning and management as illustrated for the Omo-Gibe River basin. Moreover, it can be used as an optional decision support tool to address LWFEE nexus challenges and promote climate change mitigation and delivery of river ecosystem services.” What do you mean to address LWFEE nexus challenges? You mean Gebre-model helps to address the limitation of LWFEEN?

What do you mean when you say this model in the above statement? Gebre-model or LWFEEN? I think you mean Gebre-model and is Gebre-model a kind of extension on LWFEEN that combines land and water allocation while LWFEEN was for one alone?

Would you please explain models quite well like this their connection and how independent they are etc. so that readers will be confused?

Response

Dear reviewer thank you very much for underlining this issue.

The Gebre-model is a mathematical optimization model designed to address the interactions between land, water, food, energy, and the environment (LWFEEN). The acronym LWFEEN stands for land, water, food, energy, and environment. Therefore, the LWFEEN is not a model itself, but rather a way to express the various interconnected issues. These issues are optimized by the so-called Gebre-model.

The Gebre model allocates land and water resources to meet water, food, energy, and environment conservation under the context of climate-smart land use planning(allocating crops that promote climate change mitigation) and preserving river ecosystem(allowing at least a minimum river flow along the river segments).

  • Would you please explain model components, like if LWFEEN is the main model then please have some conceptual diagram that indicate model components/sections under LWFEEN with their function?

Response

      Dear review, thank you very much for the comment.

As explained in the above sections, LWFEEN is not a model it is an acronym that stands for land-water-food-energy and environment. These issues are discussed in the introduction part. For example, land uses water to grow crop ( if there are too much rain or water sources(irrigation) so large land size can be used to grow crops; water is used by the crop (e.g., water is allocated to meet crop water requirement indirectly it is used to produce food), water produces hydropower energy (e.g. water is allocated to generate hydropower like Gibe I,II,III,IV ); agricultural productivity or land use produces environmental induced problems like soil loss, and nitrate leaching from fertilizer use).

Therefore, it is not a model, however, it is the representation of the multiple sectors' interactions.

Figure 1 is included to show the interactions between land, water, energy, and enevironment under climate change mitigation and preserving river ecosystem.

  • If there any key assumption about model water allocation from source to demand point? Like, how do the model allocate water to the demand point is that with order of demand priorities? or with percent share? or how do the model allocate water to the demand point? Otherwise, there will be water supply for the last demand point

Response.

Dear review thank you for pointing out this issue.

The Gebre-model is designed to allocate water demand to various users based on established water demand limits. For example, it takes into account the water intake required for hydropower generation (such as for Gibe I, II, III, IV) which has an average minimum and maximum demand limit per week. The model simulates the allocation of water within this range over a specified period of time. If the water is not allocated or allocated below the minimum demand, the model incorporates a penalty cost.

In a similar fashion, the Gebre-model considers the water demand limit for each individual demand node. It optimizes the limited water resources to meet the various water demands, such as those for hydropower generation and water supply for towns.

8)Is there any way that you can present statement about model uncertainty? how is your result reliable?

Response

Dear review, thank you very much for the comment.

With regards to model result uncertainty assessment, there are a few different ways that this can be done. One approach is to use statistical measures such as to evaluate the results with observed data.

In this case, the output with regard to the water allocation is based on the hydropower water demand flow limit(which is based on the observed data), so the target is to meet those observed demand flows. The other issue is the river flow, and volume is also based on the observed data obtained from the Ethiopian Ministry of water and energy. The river flow and volume are assumed based on the collected observed data along the river segment, including the tributaries inflow to the main river (e.g.,Wabi, Gilgel gibe, Gojeb, and Weybo river).

With regard to uncertainty, we chose to evaluate the model using another approach to perform sensitivity analysis to explore how the model's results change in response to variations in the input variables. This is indirectly called model scenario-based sensitivity analysis(Page 14-17). This has been done under different scenario conditions and the model has responded to the changes in put parameters.

The limitation and uncertainty that arises from the data use, other model structure or framework, and so on are explained in the manuscript as a limitation and future research challenge. In summary, we aimed to develop a model that addresses different complex interactions to produce an optimal solution. (please see the discussion and conclusion part).

To elaborate, it is well discussed in the discussion part to evaluate the model under different scenario-based sensitivity analyses.

Dear reviewer,

We are grateful for your efforts in reviewing our manuscript. Your attention to detail and insightful suggestions have greatly enhanced the quality of our work. Your contributions are deeply appreciated and have improved the manuscript greatly. The manuscript is revised, and the confusing texts and spacing problems are resolved carefully. Further, the manuscript is thoroughly revised and improved language to easily readers understand the findings.

Once again, thank you for your invaluable support.

Reviewer 2 Report

The reviewed article correctly describes optimizing the combined allocation of land and water to agriculture in the Omo-Gibe river basin considering the water-energy-food-nexus and environmental constraints. The literature is appropriately selected for the discussed issues. The tabular elements contain all the required information.

1. Are the thousands of pictures in the publication your own?

2. Where did the idea for such research come from?

3. Resolution eg Figure 15-33 is to be improved.

4. Does the analyzed problem also occur in a significant way in other parts of the world?

5. The literature needs to be unified, because the same rules are not maintained. For example, compare the notation of items 44 and 51.

Author Response

Response to reviewer2 Comments and Suggestions

The reviewed article correctly describes optimizing the combined allocation of land and water to agriculture in the Omo-Gibe river basin considering the water-energy-food-nexus and environmental constraints. The literature is appropriately selected for the discussed issues. The tabular elements contain all the required information.

  1. Are the thousands of pictures in the publication your own?

Response

Dear reviewer thank you for the comment.

All pictures are part of the research and are the authors` own analysis results.

  1. Where did the idea for such research come from?

Response

Dear reviewer thank you for the comment.

The idea arises from a research gap. We had two publications that deal with land and water allocation separately. We review publications between 2000-2019, we found that there is a lack of decision tools that solve combined land and water allocation. So we started formulating and developing an optimization model that fills the research gap.

1.Gebre, D. Cattrysse, E. Alemayehu, and J. van Orshoven. “Multi-criteria decision making methods to address rural land allocation problems: A systematic review,” International Soil and Water Conservation Research, vol. 9, no. 4, pp. 490–501, Dec. 2021, doi: 10.1016/j.iswcr.2021.04.005.

  1. Gebre, S.L.; Cattrysse, D.; Van Orshoven. “Multi-criteria decision-making methods to address water allocation problems: A systematic review,” Water, vol. 13, no. 125, 2021.
  2. Resolution e.g. Figure 15-33 is to be improved.

Response

Dear reviewer thank you for the comment.

The figures' resolution is improved. Some of the figures were squeezed due to space but are now stretched.

  1. Does the analyzed problem also occur in a significant way in other parts of the world?

Response

Dear reviewer thank you for the comment.

The use of the model to manage the interplay of land, water, food, energy, and environment is crucial in fulfilling the increasing demand for ecosystem services. Resource allocation issues between various uses and users exist in many river basins worldwide, such as the Ganges basin in India and Bangladesh, the Huai basin in China, the Rhine basin in Europe, and more.

Rising population and economic development, combined with changes in consumption patterns, are increasing the demand for water, food, energy, and environmental conservation. This places a strain on limited natural resources, making efficient resource allocation critical for sustainability.

The issue of land and water resource allocation is thoroughly analyzed in our review for identifying future research challenges. This problem affects various regions globally and requires further research to address the issues and preserve natural resources.

  1. The literature needs to be unified, because the same rules are not maintained. For example, compare the notation of items 44 and 51.

Response

Dear reviewer thank you for the comment.

Corrected throughout the manuscript.

All the literature reference styles are adjusted according to the journal style.

Dear reviewer,

We would like to express our sincere gratitude for taking the time to review our manuscript. Your valuable insights and recommendations have been instrumental in improving the quality of our work. Your contribution to this process is greatly appreciated and has been invaluable to us.

Thank you once again for your support and expertise. We value your contribution to the peer-review process and appreciate your efforts in making it.

Sincerely,

author
